# PRMT5-mediated arginine methylation activates AKT kinase to govern tumorigenesis

Shasha Yin[1,4], Liu Liu[1,4], Charles Brobbey[1], Viswanathan Palanisamy [1], Lauren E. Ball[2], Shaun K. Olsen [3], Michael C. Ostrowski [1] & Wenjian Gan [1✉]

AKT is involved in a number of key cellular processes including cell proliferation, apoptosis and metabolism. Hyperactivation of AKT is associated with many pathological conditions, particularly cancers. Emerging evidence indicates that arginine methylation is involved in modulating AKT signaling pathway. However, whether and how arginine methylation directly regulates AKT kinase activity remain unknown. Here we report that protein arginine methyltransferase 5 (PRMT5), but not other PRMTs, promotes AKT activation by catalyzing symmetric dimethylation of AKT1 at arginine 391 (R391). Mechanistically, AKT1-R391 methylation cooperates with phosphatidylinositol 3,4,5 trisphosphate (PIP3) to relieve the pleckstrin homology (PH)-in conformation, leading to AKT1 membrane translocation and subsequent activation by phosphoinositide-dependent kinase-1 (PDK1) and the mechanistic target of rapamycin complex 2 (mTORC2). As a result, deficiency in AKT1-R391 methylation significantly suppresses AKT1 kinase activity and tumorigenesis. Lastly, we show that PRMT5 inhibitor synergizes with AKT inhibitor or chemotherapeutic drugs to enhance cell death. Altogether, our study suggests that R391 methylation is an important step for AKT activation and its oncogenic function.

[1] Department of Biochemistry and Molecular Biology, Hollings Cancer Center, Medical University of South Carolina, Charleston, SC, USA. [2] Department of Cell and Molecular Pharmacology, and Experimental Therapeutics, Medical University of South Carolina, Charleston, SC, USA. [3] Department of Biochemistry and Structural Biology, University of Texas Health Science Center at San Antonio, San Antonio, TX, USA. [4] These authors contributed equally: Shasha Yin, Liu Liu. ✉email: ganw@musc.edu

The AKT/PKB protein kinase family, including AKT1, AKT2, and AKT3, plays a key role in cell proliferation, survival, metabolism, genome stability, and tissue invasion[1,2]. Hyperactivation of AKT is observed in various human cancers including breast cancer[3–5]. Moreover, a number of studies have demonstrated that aberrant activation of AKT signaling promotes tumorigenesis in mouse models[6–9]. Therefore, AKT is an ideal therapeutic target.

AKT activation is tightly controlled through multiple steps in response to growth factor signaling. Specifically, activation of class I phosphoinositide 3 kinase (PI3K) coverts phosphatidylinositol (4,5)-bisphosphate (PIP2) to PIP3, which binds AKT-PH domain to induce cytosolic AKT membrane translocation and conformational changes[10–12]. Subsequently, PDK1 and mTORC2 phosphorylate AKT at Thr308 (pT308) and Ser473 (pS473), respectively[13,14], leading to full activation of AKT and phosphorylation of many substrates to exert its oncogenic functions[2]. In addition to phosphorylation, other posttranslational modifications (PTMs) including ubiquitination and methylation have been reported to regulate AKT activation[15].

Recently, protein arginine methylation has been identified as one of the most common PTMs[16], which governs enzymatic activity, protein–protein interaction, protein localization, and stability[17,18]. Protein arginine methyltransferases (PRMTs) are responsible for transferring the methyl group to the guanidino nitrogen atoms of arginine, forming three types of methylated arginine residue: monomethylarginine (MMA), asymmetric dimethylarginine (aDMA), and symmetric dimethylarginine (sDMA)[19]. In mammals, nine PRMT family members have been identified, which are classified into three distinct types according to their catalytic activity. Type I enzymes (PRMT1, 2, 3, 4, 6, and 8) catalyze the formation of aDMA, whereas type II enzymes (PRMT5 and PRMT9) promote sDMA. Type III enzyme (PRMT7) only generates MMA. Notably, PRMT1 is the main type I enzyme responsible for about 80% of total arginine methylation, whereas PRMT5 is the dominant type II enzyme[20]. These PRMTs are involved in various fundamental cellular processes including DNA replication, transcription, RNA processing, DNA repair, and protein degradation[21–23]. Deregulation of PRMTs is associated with human diseases including cancers[24,25].

Emerging evidence showed that PRMT5 is involved in regulating activation of the AKT signaling pathway. Depletion of PRMT5 impaired AKT phosphorylation and suppressed lung cancer cell proliferation[26]. Moreover, AKT phosphorylation was decreased in purified hematopoietic stem and progenitor cells isolated from Prmt5-null mice[27]. However, the precise mechanism by which PRMT5 controls AKT activation is unknown. Here we reported that PRMT5 directly methylates AKT1-R391 and cooperates with PI3K to promote AKT kinase activity and its oncogenic function.

## Results
**PRMT5 is required for AKT activation.** To determine whether PRMTs are involved in regulation of AKT activation, we performed a CRISPR-Cas9-based screen to knock out individual PRMT gene in MCF7 breast cancer cells. PRMT8 was excluded in this screen, because its expression is restricted to the brain[28]. Notably, knockout of PRMT5, but not other PRMTs, markedly blocked phosphorylation of AKT and its downstream substrates including glycogen synthase kinase-3β (GSK-3β) and forkhead box class O 3a (FOXO3A) (Fig. 1a and Supplementary Fig. 1a–g). This observation was confirmed in other breast cancer cells including MDA-MB-231, T-47D, and MDA-MB-436, colon cancer cells DLD-1, and cervical cancer cells HeLa (Supplementary Fig. 1h–l), supporting that PRMT5-mediated AKT activation

is unlikely to be cell-type dependent. Consistently, knockdown of PRMT5 by short hairpin RNA (shRNA) or pharmacological inhibition of PRMT5 by its specific inhibitor GSK591[29] led to the inactivation of AKT (Fig. 1b, c). Importantly, AKT1 immunopurified from PRMT5-knockout cells exhibited much lower kinase activity than AKT1 immunopurified from control cells in the in vitro kinase assay with GSK-3β as a substrate (Fig. 1d). Moreover, AKT activation was also largely abrogated in PRMT5-depleted MCF7 cells in response to insulin stimulation (Fig. 1e). To determine whether PRMT5 also regulates the activation of AKT2 and AKT3, we generated AKT1/2-KO and AKT1/3-KO in MDA-MB-231 cells that express all the isoforms of AKT (Supplementary Fig. 1m, n). Notably, knockout of PRMT5 in these cells also reduced AKT phosphorylation, suggesting impaired activation of AKT2 and AKT3 (Supplementary Fig. 1o, p). Altogether, these results indicate a direct link between AKT kinase activity and PRMT5.

**AKT is a critical downstream effector of PRMT5 on cell proliferation and tumor growth.** As the main type II enzyme, PRMT5 mostly functions as an oncoprotein by governing cellular proliferative/apoptotic signaling[30]. In support of this notion, we found that depletion of endogenous PRMT5 led to reduced colony formation and anchorage-independent cell growth (Supplementary Fig. 2a, b). Moreover, re-introduction of PRMT5-WT, but not the PRMT5-E444Q mutant (the enzymatically dead form of PRMT5), could rescue AKT activation and cell proliferation of PRMT5-depleted cells (Supplementary Fig. 2c–e), suggesting that the methyltransferase activity is required for the proliferative function of PRMT5. Given that AKT is a master regulator of cell proliferation, we investigated whether AKT is a downstream effector of PRMT5. Notably, ectopic expression of the constitutively active form of AKT1 (myr-AKT1) largely rescued AKT signaling and the colony-forming ability of PRMT5-depleted MCF7 cells (Fig. 2a–c). Moreover, knockdown of PRMT5 significantly suppressed tumor growth, which could be reversed by myr-AKT1 (Fig. 2d, e and Supplementary Fig. 2f). In keeping with these observations, immunohistochemistry (IHC) analysis of Ki-67 showed decreased proliferating cells in PRMT5-depleted tumors, which could be rescued by expressing myr-AKT1 (Fig. 2f). Consistently, ectopic expression of myr-AKT1 also suppressed cell apoptosis in PRMT5-depleted tumors as evidenced by a significant reduction of TUNEL (terminal deoxynucleotidyl transferase dUTP nick end labeling)-stained cells (Fig. 2g, h). These results collectively demonstrate that PRMT5 exerts its oncogenic function in part through activating the AKT pathway.

**PRMT5 is the arginine methyltransferase of AKT1.** Next, we sought to investigate whether AKT is a bona fide substrate of PRMT5. To this end, we found that PRMT5 specifically co-immunoprecipitated AKT1, but neither S6K1 nor PDK1, another two members of the AGC kinase family[31] (Fig. 3a and Supplementary Fig. 3a). Reciprocally, AKT1 interacted with PRMT5 but not PRMT1 (Fig. 3b). Moreover, PRMT5 bound all isoforms of AKT including AKT1, AKT2, and, to a lesser extent, AKT3 (Supplementary Fig. 3b). We further mapped which domain of AKT1 interacts with PRMT5. AKT1 consists of an N-terminal PH domain, a middle kinase domain (KD), and a C-terminal regulatory region (Tail)[2]. We found that PRMT5 bound to the AKT-KD and, to a lesser extent, the PH domain (Supplementary Fig. 3c, d). Consistent with these binding results, PRMT5, but not PRMT1 nor PRMT9, could catalyze the sDMA formation on AKT1, but not S6K1, in an enzymatic activity-dependent manner (Supplementary Fig. 3e–g). On the other hand, depletion of

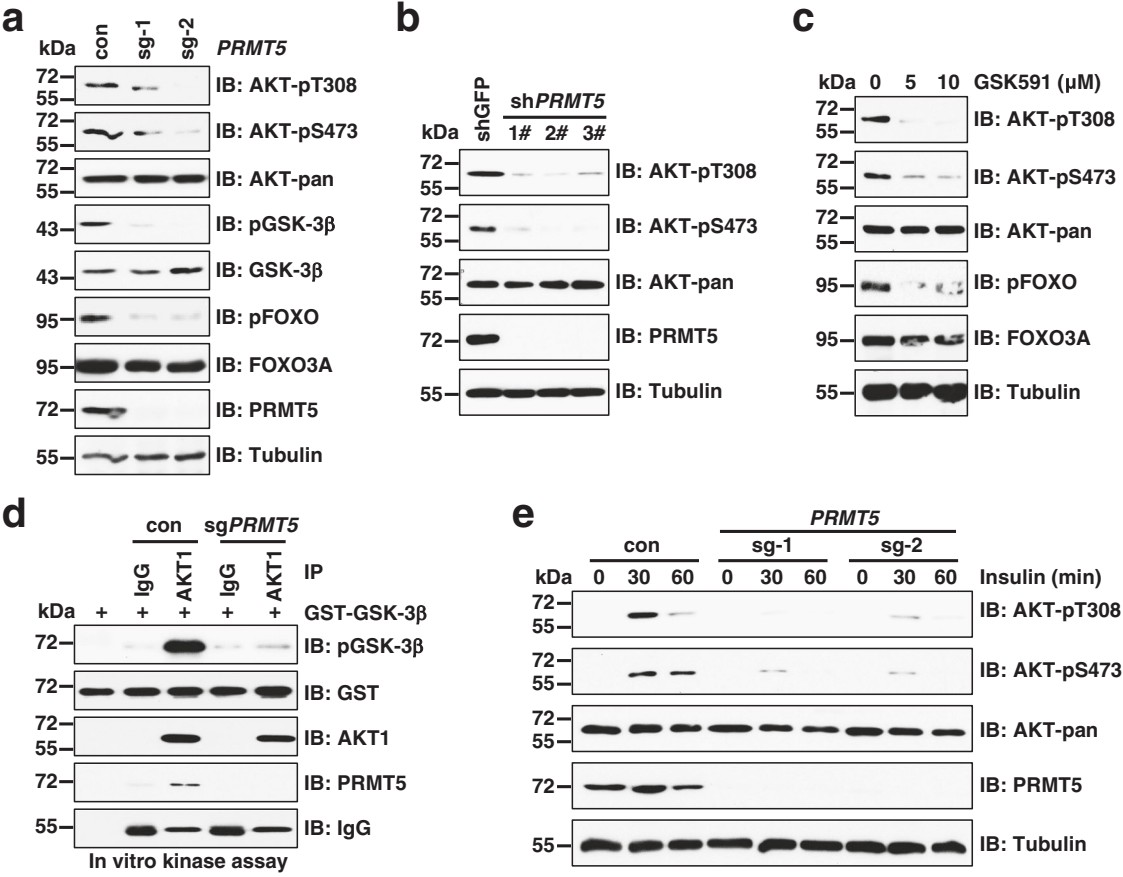

**Fig. 1 Deficiency in PRMT5 suppresses AKT activation. a, b** Immunoblot (IB) analysis of whole cell lysates (WCLs) derived from MCF7 cells infected with lentiCRISPR virus (**a**) or shRNAs virus targeting *PRMT5* (**b**). The cells were selected with 2 μg/ml puromycin for 4 days to eliminate the non-infected cells. **c** IB analysis of WCLs derived from MCF7 cells treated with GSK591 for 24 h. **d** AKT in vitro kinase assays were performed using endogenous AKT1 (IgG as a negative control) immunopurified (IP) from control cells or sg*PRMT5* cells as the kinase source and recombinant GST-GSK-3β purified from bacteria as the substrate. **e** IB analysis of WCLs derived from control cells or *PRMT5*-depleted MCF7 cells. Cells were serum-starved for 16 h and then treated with 100 nM insulin for 0, 30, and 60 min before collecting. Similar results were obtained in *n* ≥ 2 independent experiments in **a–e**. Uncropped immunoblots are provided in Source Data file.

*PRMT5* blocked AKT1 sDMA formation (Supplementary Fig. 3h). Importantly, in vitro methylation assays demonstrated that PRMT5 directly methylates AKT1 in a methyltransferase activity-dependent manner (Fig. 3c).

To identify the arginine residue(s) that is methylated by PRMT5, we analyzed the protein sequence of AKT1 using methylation prediction tools including PRmePred[32] and GPS-MSP[33]. Six arginine residues ranked top scores were selected for further analyses (Supplementary Fig. 3i). Interestingly, the substitution of R391 to K (R391K) completely blocked AKT1 sDMA formation in cells (Supplementary Fig. 3j). Of note, a recent study on bioRxiv showed that AKT1-R15 is methylated in neuroblastoma cells (https://doi.org/10.1101/2020.08.12.246660). However, we found that R15K mutation fails to inhibit AKT1 sDMA signal in our cell lines (Supplementary Fig. 3j, k), indicating the site of methylation in AKT1 possibly occurs in a cell-type-specific manner. Moreover, high-resolution liquid chromatography with tandem mass spectrometry (MS/MS) analyses confirmed dimethylation of AKT1-R391 in our cells (Supplementary Fig. 3l). Notably, AKT1-R391 is evolutionarily conserved from *Drosophila* to *Homo sapiens* (Fig. 3d), indicating a potential role of this methylation event in various species.

To further confirm AKT1-R391 methylation, we generated a polyclonal antibody that specifically recognizes symmetric dimethyl R391 (R391-me2s), but not the unmodified, mono-methyl, or asymmetric dimethyl R391 as assessed in the dot blot assays (Supplementary Fig. 3m). R391K mutation completely blocked PRMT5-mediated methylation of AKT1 in vitro and in cells (Fig. 3e, f and Supplementary Fig. 3n). Moreover, AKT1-R391-me2s was abolished in *PRMT5*-knockout cells (Fig. 3g). Consistently, AKT2 and AKT3 were also methylated at corresponding residues, mutating of which blocked the R391-me2s signal (Supplementary Fig. 3o, p). Lastly, we found that the expression levels of PRMT5 is positively correlated with AKT1-R391-me2s signal in breast cancer patient specimens (Fig. 3h, i). Altogether, these results suggest that PRMT5 is the physiological arginine methyltransferase responsible for the symmetric methylation of AKT1-R391.

**PI3K signaling promotes AKT1-R391 methylation.** It is well established that in response to growth factor stimulation, such as insulin or epidermal growth factor (EGF), activated PI3K coverts PIP2 to PIP3, which binds the AKT-PH domain and leads to AKT conformational change[10–12]. Thus, we investigated the crosstalk between PRMT5-mediated AKT methylation and the PI3K signaling. Interestingly, insulin or EGF stimulation enhanced the interaction of AKT1 with PRMT5 and subsequently promoted AKT1-R391-me2s, coincided with AKT phosphorylation (Fig. 4a, b and Supplementary Fig. 4a–c). However, growth

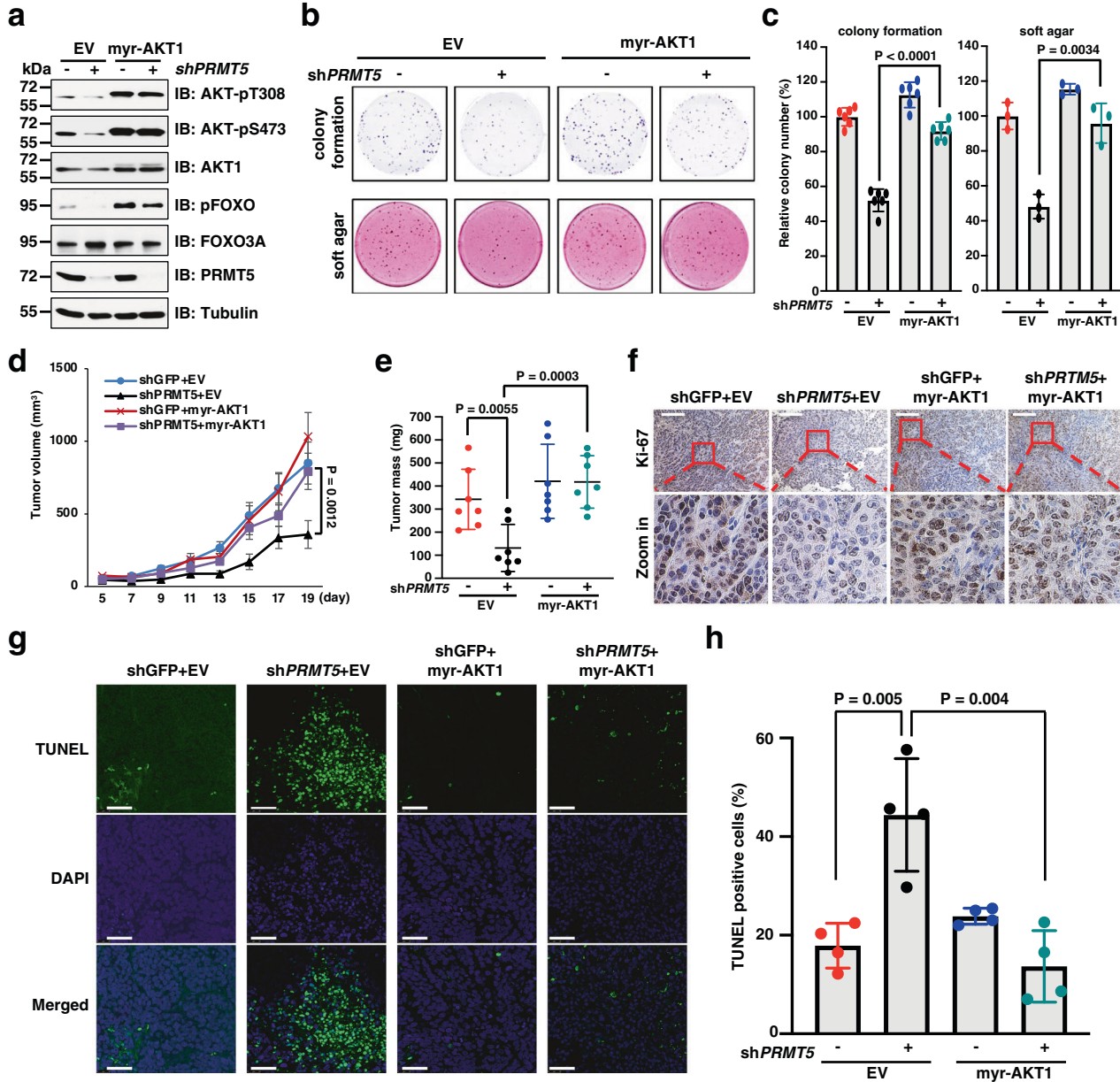

**Fig. 2 AKT rescues cell proliferation and tumor growth of *PRMT5*-depleted cells. a** MCF7 cells stably expressing myr-AKT1 were infected with shRNA targeting *PRMT5* and subjected to IB analysis. Similar results were obtained in $n = 3$ independent experiments. **b**, **c** Cells generated in **a** were subjected to colony formation and soft agar assays. Representative images are shown in **b** and relative colony numbers are plotted in **c**. Data are shown as the mean ± SD of $n = 3$ or 6 independent experiments. **d** DLD-1 cells ectopically expressing myr-AKT1 and/or depleting *PRMT5* were injected into nude mice. Tumor growth was monitored for the indicated time period. Data are shown as the mean ± SEM for $n = 7$ mice. **e** Weight of the tumors. Data are shown as the mean ± SD for $n = 7$ tumors. **f** Representative images of IHC staining of Ki-67 in $n = 4$ xenograft tumors. Scale bar, 100 μm. **g** Representative images of TUNEL assays in xenograft tumors. Scale bar, 50 μm. **h** Quantification of TUNEL-positive cells in xenograft tumors. Data are shown as the mean ± SD for $n = 4$ tumors. Statistical significance was determined by two-tailed Student's *t*-test in **c**, **e**, **h** and by two-way ANOVA in **d**. Uncropped immunoblots and statistical source data are provided in Source Data files.

factors have minimal effect on H4R3-me2s, indicating growth factors may not affect PRMT5 methyltransferase activity (Supplementary Fig. 4d, e). Interestingly, treatment of PI3K inhibitors reduced AKT1 interaction with PRMT5 and AKT1-R391 methylation (Fig. 4c), indicating that PI3K is an upstream regulator of AKT1-R391 methylation. Further supporting this notion, the non-PIP3-binding mutation AKT1-R25C diminished AKT1 interaction with PRMT5, leading to a reduction of AKT1-R391 methylation (Fig. 4d, e). In contrast, the patient-derived AKT1-E17K mutation[34], which enhances the affinity for PIP2/3, promoted AKT1 interaction with PRMT5 and AKT1-R391

methylation (Supplementary Fig. 4f, g). Importantly, PIP3 enhanced PRMT5 binding to AKT1-WT, but not AKT1-R25C in vitro (Fig. 4f). These results suggest that PI3K signaling may enhance PRMT5-mediated AKT1-R391 methylation through PIP3-induced AKT confirmational change.

**AKT1-R391 methylation plays a critical role in AKT activation processes**. We next sought to explore whether, on the other hand, PRMT5-mediated R391 methylation plays a role in PI3K-dependent AKT activation processes. Previous studies have illustrated that without the growth factor stimulation, intramolecular

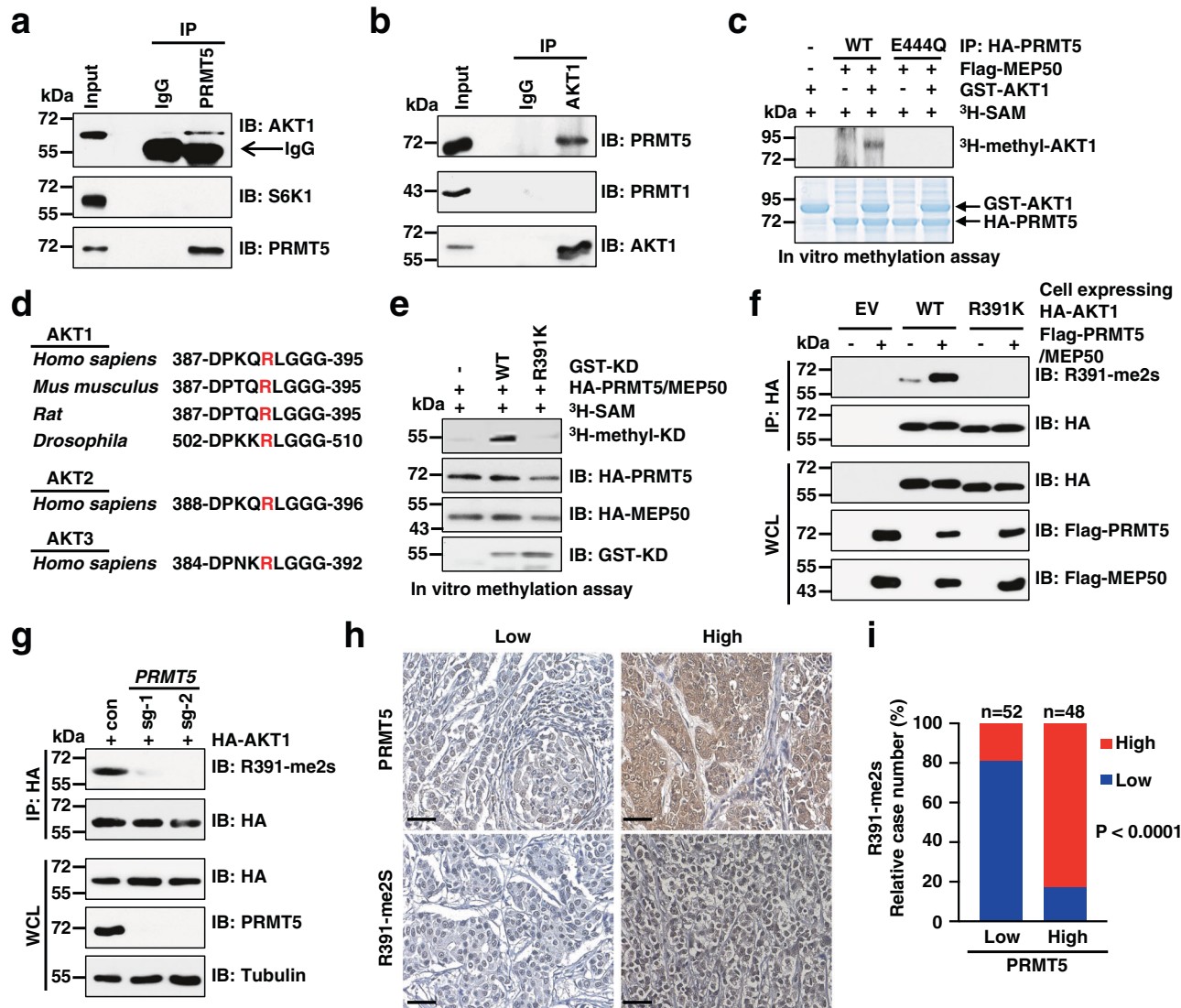

**Fig. 3 PRMT5 catalyzes symmetric dimethylation of AKT1 at R391. a, b** IB analysis of WCLs and immunoprecipitation (IP) products derived from MCF7 cells. IgG was used as a negative control. **c** In vitro methylation of AKT1 in the presence of ³H-SAM. GST-AKT1, PRMT5-WT, and PRMT5-E444Q proteins were purified from HEK293 cells. **d** Sequence of the evolutionarily conserved residue R391 (red) in AKT. **e** In vitro methylation of AKT1-KD-WT and AKT1-KD-R391K in the presence of ³H-SAM. Recombinant GST-AKT1-WT and AKT1-KD-R391K proteins were purified from bacteria and HA-PRMT5/MEP50 proteins were immunopurified from HEK293 cells. **f** IB analysis of WCLs and IP products derived from MCF7 cells stably expressing AKT1-WT or R391K. Flag-PRMT5/MEP50 was transiently transfected into these cells. **g** Knockout of *PRMT5* abrogates AKT1-R391 methylation. IB analysis of WCLs and IP products derived from *PRMT5*-knockout cells or control cells transfected with HA-AKT1. **h** Representative images of IHC staining for PRMT5 and R391-me2s in a breast cancer tissue array (n = 100 tissue specimens). Scale bar, 50 μm. **i** Quantification of cases with PRMT5 and R391-me2s staining (n = 100 tissue specimens). P-values were calculated using two-tailed $\chi^2$-test. Similar results were obtained in n ≥ 2 independent experiments in **a–c** and **e–g**. Uncropped immunoblots and statistical source data are provided in Source Data files.

PH domain and KD domain interactions maintain AKT in a closed conformation, called PH-in[35,36]. Notably, AKT1-R391K mutation or depletion of *PRMT5* strengthened the PH-KD interaction (Fig. 5a and Supplementary Fig. 5a), and, inversely, ectopic expression of PRMT5 decreased KD-WT, but not the KD-R391K mutant, interaction with the PH domain (Supplementary Fig. 5b). Interestingly, PRMT5 also perturbed KD interaction with PH-R25C mutant (Supplementary Fig. 5c), indicating that PRMT5 promotes AKT confirmation change independent of its binding to PIP3. Importantly, the symmetric dimethyl R391 peptide displayed a much lower PH-binding capacity than the corresponding non-modified peptide (Fig. 5b). Moreover, abrogation of AKT methylation through R391K mutation or *PRMT5* depletion diminished the interaction between AKT1 and PIP3

(Fig. 5c, d), and consequently reduced AKT1 membrane recruitment (Fig. 5e, f and Supplementary Fig. 5d–f). These data suggest that PRMT5-mediated R391 methylation may weaken intramolecular PH-KD binding to facilitate AKT conformational shift from the PH-in to PH-out state and membrane translocation.

To gain further insights into the molecular mechanism by which methylation of AKT1-R391 regulates phosphorylation of AKT, we found that the interaction between AKT1 and PDK1 was decreased in AKT-R391 methylation-deficient cells (Fig. 5g, h and Supplementary Fig. 6a, b), which mimics the behavior of the AKT1-R25C mutant (Supplementary Fig. 6c, d). On the other hand, overexpression of PRMT5-WT, but not the PRMT5-E444Q mutant, promoted AKT1 interaction with PDK1 (Supplementary

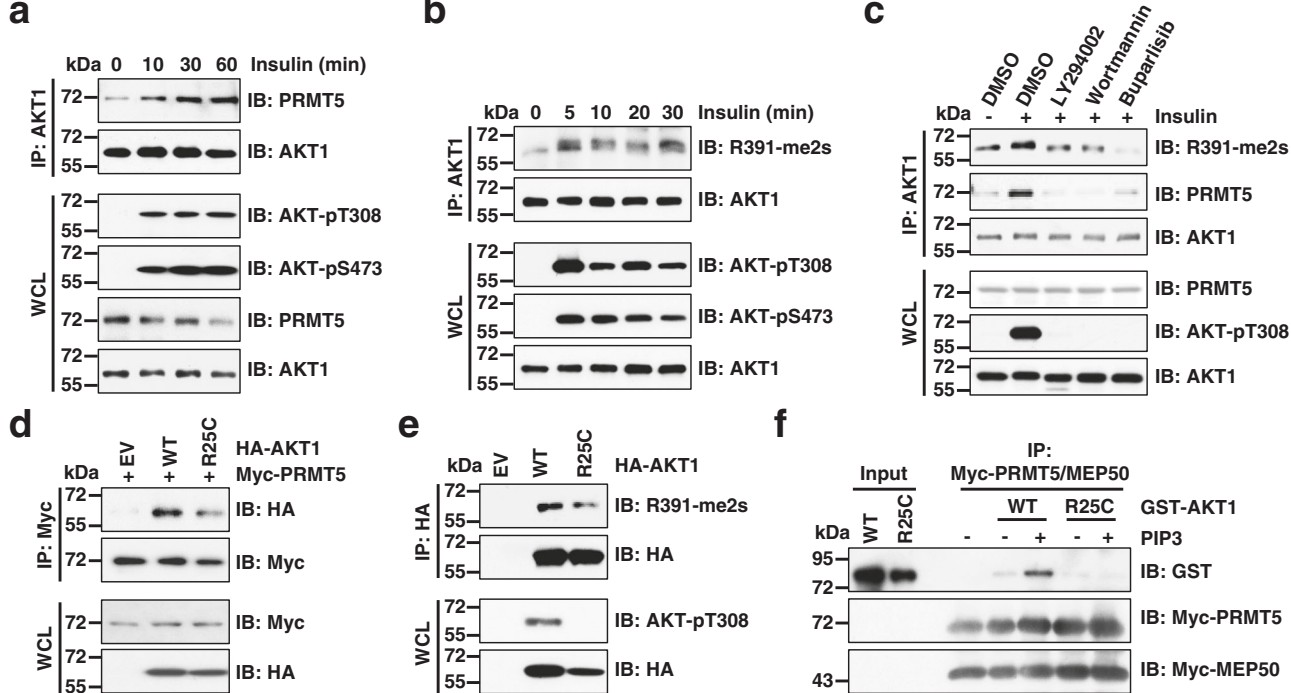

**Fig. 4 PI3K signaling enhances AKT1-R391 dimethylation. a, b** IB analysis of WCLs and IP products derived from MCF7 cells. The cells were serum-starved for 16 h and then treated with 100 nM insulin for indicated time periods before collection. **c** IB analysis of WCLs and IP products derived from MCF7 cells. The cells were serum-starved for 24 h and treated with PI3K inhibitors for 12 h followed by insulin for 30 min before collection. LY294002 (20 µM), Wortmannin (10 µM), and Buparlisib (10 µM). **d, e** IB analysis of WCLs and IP products derived from HEK293T cells transfected with indicated constructs. **f** In vitro binding assays were performed with recombinant GST-AKT1 protein purified from mammalian cells and Myc beads bound with PRMT5/MEP50. The binding was performed at 4 °C for 4 h incubated with or without PIP3 (20 µM) and subjected to IB analysis. Similar results were obtained in $n \geq 2$ independent experiments in **a–f**. Uncropped immunoblots are provided in Source Data file.

Fig. 6e). We also observed that AKT1-R391K mutation or depletion of *PRMT5* impairs the interaction between AKT1 and Sin1 (Supplementary Fig. 6f, g), the subunit of mTORC2 complex mediating its association with substrate[37]. These results suggest that AKT-R391 methylation enhances AKT interaction with its upstream kinases, PDK1 and mTORC2.

Lastly, analysis of the AKT1 structure showed that R391 participates in a conserved network of salt bridges and hydrogen bonds with E319, A329, and K386 (Supplementary Fig. 6h). Notably, this serves to position a loop that harbors the conserved "DYG" motif, which participates in a network of interactions with the AKT1 activation loop and the pT308 side chain (PDB: 3CQW)[38], and serves as a crucial regulatory region of many kinases[39]. Modeling of symmetrically dimethylated R391 onto AKT1 showed that this network of interactions is disrupted both due to a loss of hydrogen bond donors on the dimethylated R391 side chain as well as the steric clashes that occur (Supplementary Fig. 6i). Given the connection of this region to the DYG motif and activation loop, which includes T308, we speculate that dimethylation of R391 may result in a promotion of T308 phosphorylation via an allosteric effect.

**Deficiency in AKT1-R391 methylation suppresses AKT activation and oncogenic function.** Given that AKT1-R391 methylation is critical for AKT signaling activation, we next investigated whether it governs AKT biological functions. To this end, we re-constituted AKT1-WT and the methylation-deficient mutant of AKT1 (R391K) in DLD-1 cells, in which both *AKT1* and *AKT2* were knocked out (DLD-1-*AKT1/2$^{-/-}$*). Compared to AKT1-WT, the AKT1-R391K mutation significantly reduced AKT

activation at the basal level and in response to stimulation by growth factors (Fig. 6a, b and Supplementary Fig. 7a). Consistently, the kinase activity of R391K was dramatically decreased as shown by in vitro kinase assay with GSK-3β as a substrate (Fig. 6c). We also observed similar phenotypes in MCF7 cells (Supplementary Fig. 7b–e). Moreover, AKT-pS473 was dramatically enriched at the membrane in cells expressing AKT1-WT, but not the R391K mutant (Supplementary Fig. 7f). Previous studies demonstrated that TSC1–TSC2 complex and oncogenic Ras mutant directly associate with mTORC2 to promote its kinase activity[40,41]. Interestingly, PRMT5 also regulated TSC2 or oncogenic Ras-mediated phosphorylation of AKT-S473 (Supplementary Fig. 7g, h), probably through modulating AKT interaction with mTORC2 (Supplementary Fig. 6f, g).

In keeping with the much lower kinase activity of R391K mutant, *AKT1/2*-KO cells expressing this mutated protein displayed a dramatic reduction in cell proliferation, colony formation, and anchorage-independent cell growth, compared to *AKT1/2*-KO cells expressing AKT1-WT protein (Fig. 6d–f and Supplementary Fig. 8a–h). Consistent with the minor effect on AKT1 arginine methylation, R15K mutation did not disturb AKT activation and its proliferative function (Supplementary Fig. 8i–k).

To pinpoint the potential role of AKT-R391 methylation in tumorigenesis in vivo, we performed xenograft mouse assays. Notably, compared to AKT1-WT, AKT1-R391K mutation significantly retarded tumor growth (Fig. 6g–i). TUNEL staining and IHC analysis of Ki-67 demonstrated an elevation of cell apoptosis and a decrease of proliferating cells in tumors expressing AKT1-R391K, compared to tumors expressing AKT1-WT (Fig. 6j–l). Importantly, depletion of *PRMT5*

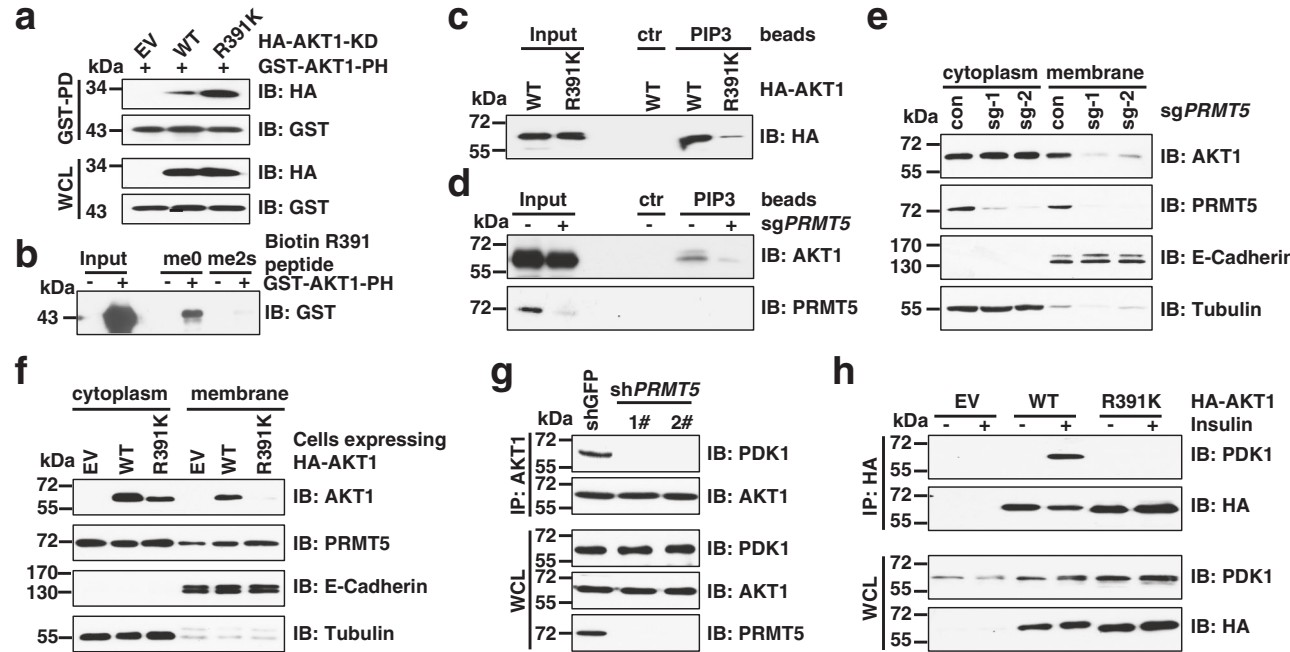

**Fig. 5 AKT1-R391 methylation promotes AKT conformational change, membrane localization, and interaction with PDK1. a** IB analysis of WCLs and GST pulldown products derived from HEK293 cells transfected with indicated constructs. **b** Biotin-conjugated R391 peptides with (me2s) or without (me0) dimethylation were incubated with recombinant GST-AKT1-PH and precipitated with streptavidin. Peptide (4 µg) was used for each sample. **c** IB analyses of PIP3 pulldown products. HA-AKT1 proteins were purified from HEK293T cells transfected with HA-AKT1-WT or HA-AKT1-R391K. Empty beads (ctr) serve as a negative control. **d** IB analyses of PIP3 pulldown products and WCLs derived from *PRMT5*-knockout MCF7 cells. **e**, **f** IB analysis of cell fractionations derived from *PRMT5*-depleted MCF7 cells (**e**) or from DLD-1-*AKT1/2⁻/⁻* cells expressing AKT1-WT or R391K mutant (**f**). **g** IB analysis of WCLs and IP products derived from *PRMT5*-knockdown MCF7 cells. **h** IB analysis of WCLs and IP products derived from DLD-1-*AKT1/2⁻/⁻* cells expressing AKT1-WT or R391K mutant. Cells were serum-starved for 16 h and then treated with 100 nM insulin for 60 min before collection. Similar results were obtained in *n* ≥ 2 independent experiments in **a–h**. Uncropped immunoblots are provided in Source Data file.

markedly suppressed AKT activation and cell proliferation in AKT1-WT cells and, to a much lesser extent, AKT1-R391K cells (Supplementary Fig. 8l, m). Altogether, these results demonstrate that PRMT5-mediated AKT-R391 methylation promotes AKT activation and its oncogenic function.

**PRMT5 inhibitor synergizes with AKT inhibitor and chemotherapeutic drugs.** PRMT5 has recently emerged as an attractive drug target for cancer therapies. Several PRMT5 inhibitors, such as GSK3326595, are on clinical trial for the treatment of a variety of cancers including breast cancer[42]. We next examine whether PRMT5 inhibitors suppresses breast cancer cell proliferation. Notably, GSK3326595 treatment significantly decreased cell survival in a time-dependent manner in a panel of breast cancer cells (Supplementary Fig. 9a–d), which is consistent with previous report[43]. Although many AKT inhibitors including AZD5363, MK2206, and GDC-0068 are in phase I/II clinical trials, none of these inhibitors has achieved satisfactory outcomes in monotherapy in part due to the inadequate AKT inhibition at tolerable doses. Therefore, a combination of AKT inhibitors with chemotherapy and other targeted therapy is a potential option for cancer treatment[44]. Interestingly, co-treatment of PRMT5 and AKT inhibitors exhibited a synergistic effect on promoting cell death in various breast cancer cells including estrogen receptor/progesterone receptor (ER/PR) positive cell lines MCF7 and T-47D, and triple-negative breast cancer cell lines MDA-MB-231, BT-549, and MDA-MB-468 (Fig. 7a, b). Previous studies also demonstrated that activation of AKT contributes to chemotherapy resistance in breast cancer[45,46]. As PRMT5 inhibitor markedly blocked AKT activation, we next examined whether PRMT5

inhibition affects cellular sensitivity to chemotherapeutic drugs. Notably, GSK3326595 sensitized breast cancer cells to etoposide and cisplatin (Fig. 7c–f). Of note, we did not observe additional effects of PRMT5 inhibitor and cisplatin in MDA-MB-468 cells, as this cell line is susceptible to cisplatin. These results suggest that targeting PRMT5 is a potential strategy to improve the cellular response to AKT inhibitors or chemotherapeutic agents.

**Discussion**

The AKT/PKB protein kinases are evolutionarily conserved from *Drosophila* to human, which are regulated by multiple upstream pathways in response to growth factor, nutrients, and hormone[2]. Hyperactivation of the AKT oncogenic pathway is a hallmark of human cancers, which can be caused by genetic alterations in AKT itself or its upstream regulators[5]. Notably, *PIK3CA* mutation or *PTEN* loss was frequently observed in a variety of human cancers[47,48], both of which lead to PIP3 accumulation and subsequent aberrant AKT activation[49,50]. Here we revealed that AKT1-R391 methylation promotes AKT kinase activity by weakening PH-KD interaction, leading to a conformational shift from the closed state to semi-open state and subsequently promoting PI3K (PIP3)-dependent membrane translocation (open state), and interaction with PDK1 and mTORC2 for phosphorylation at T308 and S473, respectively (Fig. 7g). Interestingly, this methylation event is enhanced when AKT is in an open state, suggesting a positive feedback loop between AKT-R391 methylation and AKT activation. Importantly, deficiency in AKT-R391 methylation attenuates breast tumorigenesis. Therefore, our results demonstrate that AKT1-R391 methylation is a critical step for PI3K-mediated AKT activation and oncogenic function.

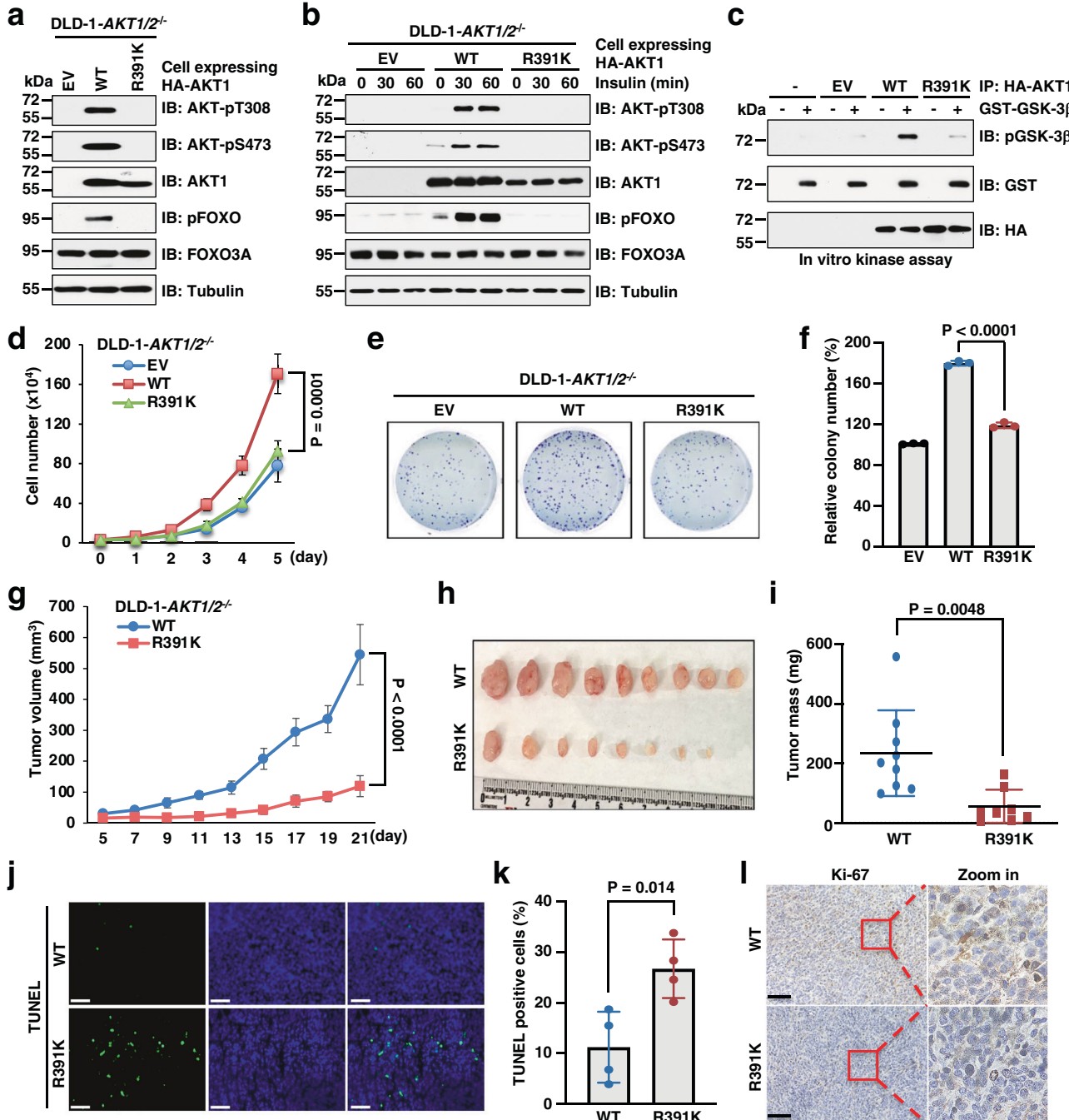

**Fig. 6 Absence of R391 methylation attenuates AKT activation and tumor growth. a, b** IB analysis of WCLs derived from DLD-1-*AKT1/2$^{-/-}$* cells expressing AKT1-WT or R391K mutant. Where indicated, cells were serum-starved for 16 h and then treated with 100 nM insulin for 0, 30, and 60 min before collection. **c** In vitro kinase assays using immunopurified HA-AKT1-WT or R391K as the kinase source and recombinant GST-GSK-3β purified from bacteria as the substrate. **d** DLD-1-*AKT1/2$^{-/-}$* cells expressing AKT1-WT or R391K mutant were subjected to cell proliferation assays. Data are shown as the mean ± SD of n = 3 independent experiments. **e** Representative images of colony formation assays. **f** Quantification of colonies. Data are shown as the mean ± SD of n = 3 independent experiments. **g-i** DLD-1-*AKT1/2$^{-/-}$* cells expressing AKT1-WT or R391K mutant were subjected to mouse xenograft assays (injection both sides of mice). Tumor growth were monitored (**g**) and dissected tumors were weighed (**h, i**). Data are shown as the mean ± SEM **g** or mean ± SD (**i**) of n = 5 mice (WT) or 4 mice (R391K). **j** Representative images of TUNEL assays in xenograft tumors. Scale bar, 50 μm. **k** Quantification of TUNEL-positive cells in xenograft tumors. Data are shown as the mean ± SD of n = 4 tumors. **l** Representative images of IHC staining of Ki-67 in n = 4 xenograft tumors. Scale bar, 100 μm. Statistical significance was determined by two-tailed Student's t-test in **f, i, k** and by two-way ANOVA in **d, g**. Similar results were obtained in n ≥ 2 independent experiments in **a–c**. Uncropped immunoblots and statistical source data are provided in Source Data files.

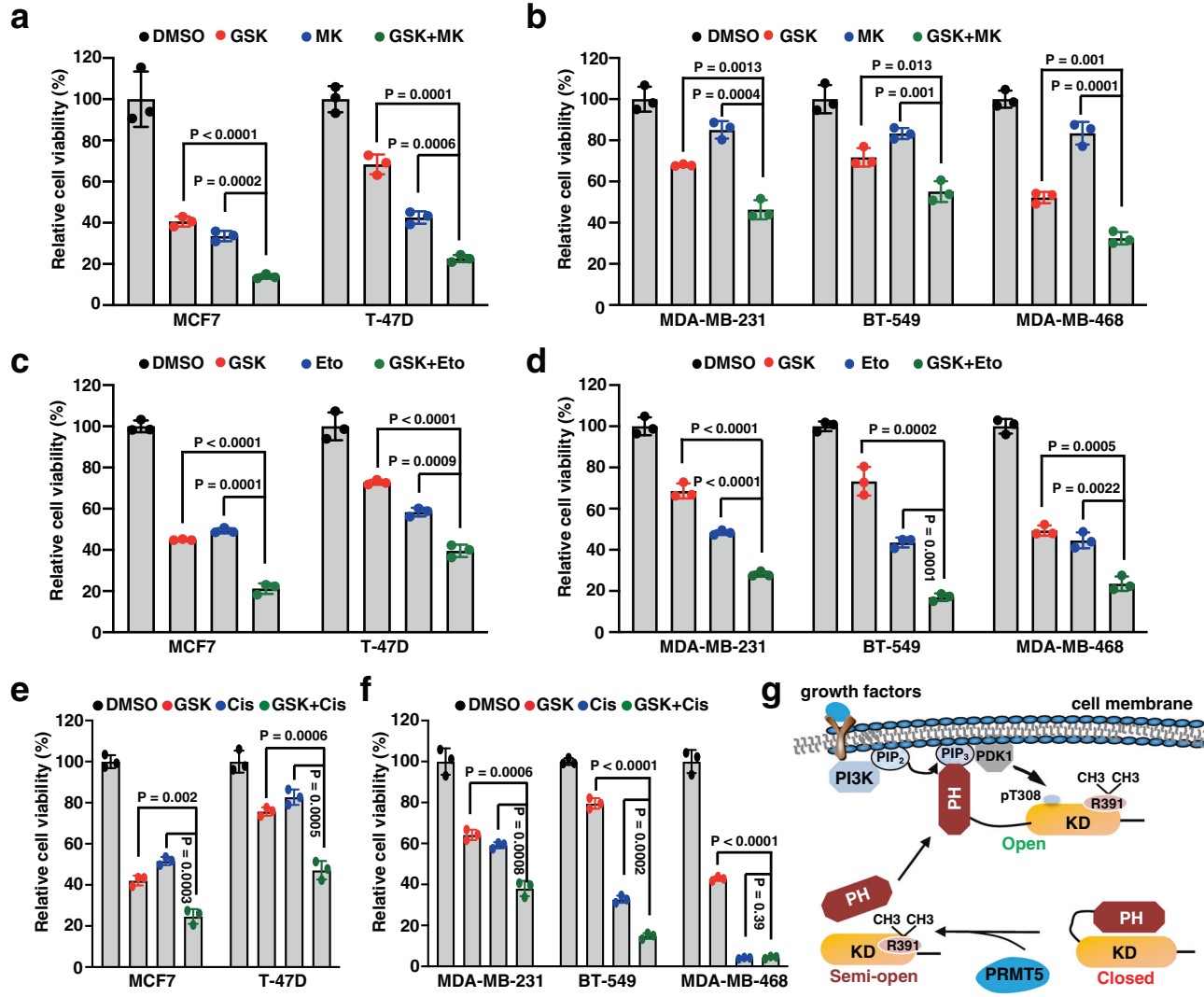

**Fig. 7 PRMT5 inhibition sensitizes breast cancer cells to AKT inhibitor and chemotherapeutic agents. a**, **b** Breast cancer cells were treated with DMSO, 0.5 µM GSK3326595 (GSK), 0.5 µM MK2206 (MK), or both (GSK + MK) for 4 days before examining cell viability. **c**, **d** Breast cancer cells were treated with DMSO, 0.5 µM GSK, 0.25 µM etoposide (Eto), or both (GSK + Eto) for 4 days before examining cell viability. **e**, **f** Breast cancer cells were treated with DMSO, 0.5 µM GSK, 2 µM cisplatin (Cis), or both (GSK + Cis) for 4 days before examining cell viability. **g** Proposed model to describe the role of AKT1-R391 methylation on promoting PI3K-dependent AKT activation. Statistical significance was determined by two-tailed Student's $t$-test in **a**–**f**. Statistical source data are provided in Source Data file.

As an epigenetic modifier of histone and non-histone proteins, PRMT5 exerts its biological functions at both cytoplasm and nucleus[51]. PRMT5 was first characterized as a transcriptional repressor by catalyzing symmetric dimethylation of histone H3 arginine 8 and H4 arginine 3, leading to transcriptional repression of tumor suppressor genes[52]. PRMT5 also regulates transcription programs by methylating transcription factors, mRNA splicing factors, and RNA-binding proteins[25]. In the cytoplasm, PRMT5 has been reported to regulate RAS-ERK and nuclear factor-κB (NF-κB) pathways by methylating epidermal growth factor receptor (EGFR) and NF-κB/p65, respectively[53,54]. Here we revealed that PRMT5 exerts its oncogenic functions, in part, by promoting the AKT pathway through directly methylating AKT1 at R391. Surprisingly, we found that a significant portion of PRMT5 is associated with the membrane, which is probably derived from Golgi membranes as previously reported[55]. Further study is necessary to classify the mechanism and function of the membrane-associated PRMT5.

PRMT5 is overexpressed in various cancers and correlated with poor prognosis, low survival, and drug resistance[56–58]. Notably, >50% of breast tumors display higher PRMT5 expression compared to normal breast tissues[56]. Moreover, overexpression of PRMT5 is associated with chemotherapeutic resistance in breast cancer[59]. Furthermore, depletion of *PRMT5* significantly inhibited breast cancer cell proliferation and tumor growth in mouse model[60]. Thus, PRMT5 is considered as a potential therapeutic target for breast cancer. Recently, several PRMT5 inhibitors have been developed including EPZ015666, GSK591, and GSK3326595[61]. Notably, the efficacy of GSK3326595 is being tested in clinical trials[42]. Our data show that PRMT5 inhibitor enhances cellular sensitivity to AKT inhibitor and chemotherapy drugs. Of note, we did not observe an apparent association between *PI3KCA* mutations and sensitivity to PRMT5 inhibitor in these breast cancer cells we examined. Further studies using more breast cancer cell lines and mouse models are required to demonstrate whether *PI3KCA* mutations dictate the efficacy of

PRMT5 inhibitor. As AKT inhibitor monotherapy had limited clinical activity in advanced breast cancer patients in part due to inadequate AKT inhibition at tolerable doses[62], our study provides a rational for the combination of PRMT5 and AKT inhibitors or chemotherapeutic drugs to achieve better clinical outcomes.

## Methods

**Cell culture, transfection, and lentivirus package.** MCF7, MDA-MB-231, HEK293, HEK293T, HeLa, DLD-1 cells, and their derived cell lines and *Tsc2* mouse embryonic fibroblasts were maintained in Dulbecco's modified Eagle's medium (DMEM) (Genesee Scientific, 25-500). T-47D, HCC1937, BT-549, MDA-MB-468, MDA-MB-436, and their derived cell lines were maintained in RPMI 1640 medium (Corning, 10-040-CV). Ten percent fetal bovine serum, 100 U/ml penicillin, and 100 μg/ml streptomycin are supplemented in the medium. Cell transfection was performed using Lipofectamine 3000 and Plus reagents following the manufacturer's instructions. Lentivirus packaging and subsequent infection of various cell lines were performed according to the protocol described previously[63]. The infected cells were selected and maintained in the presence of hygromycin (200 μg/ml), puromycin (1–2 μg/ml), or Blasticidin (10 μg/ml).

**Reagents.** Insulin (Thermo Fisher, 41400045), EGF (Thermo Fisher, PHG0311L), GSK591 (Sigma, SML1751), GSK3326595 (MedChemExpress, HY-101563), MK2206 (Selleckchem, S1078), etoposide (Sigma, E1383) cisplatin (Sigma, 232120), LY294002 (MedChemExpress, HY-10108), Wortmannin (MedChemExpress, HY-10197), and Buparlisib (MedChemExpress, HY-70063) were used at the indicated doses. PI(3,4,5)P3 beads (P-B345A-2), control beads (P-B000), and free PI(3,4,5)P3 (P-3908-2) were purchased from Echelon Biosciences.

**Cell fractionation.** This assay was performed using Subcellular Protein Fractionation Kit following the manufacturer's instructions (Thermo Fisher, 78840). The membrane fraction contains plasma, mitochondria, and ER/Golgi membranes, but not nuclear membranes.

**Plasmids.** Flag-PRMT1, Flag-PRMT5, Flag-PRMT9, and Flag-MEP50 were generated by cloning the corresponding cDNA into the pRK5-Flag vector. HA-PRMT5, HA-AKT1, and HA-AKT1-KD were constructed by cloning the corresponding cDNA into the pRK5-HA vector. Myc-PRMT5 and Myc-MEP50 were generated by cloning the corresponding cDNA into the pRK5-myc vector. pGEX-GSK-3β was generated by inserting the cDNA into pGEX-6P-1 bacteria expression vector. CMV-GST-AKT1, CMV-GST-AKT1-PH, CMV-GST-AKT1-KD, and CMV-GST-AKT1-Tail were cloned into mammalian expression pCMV-GST vector. HA-AKT2, HA-AKT3, HA-AKT1-R25C, HA-AKT1-E17K, and HA-S6K1 were described previously[64,65]. Lentiviral HA-AKT1 and PRMT5 were generated by cloning the corresponding cDNA into pLenti-HA-hygro vector or pLJM1-puro vector. HA-NRAS-WT (#120569) and myr-AKT1 (#64606) were purchased from Addgene. HA-NRAS-G12V was generated by cloning the cDNA of NRAS-G12V into the pLEX-FHH-Empty Vector (Addgene, #120568). PRMT5-E444Q and various AKT1 mutants were generated using the QuikChange XL site-directed mutagenesis kit from Agilent (#20518) according to the manufacturer's instructions. The shRNA targeting PRMT5 were purchased from Sigma (TRCN0000303446, TRCN0000303447, and TRCN0000107086). Various single guide RNAs (sgRNA) were designed at https://www.synthego.com and were cloned into lentiCRISPR v2 vector (Addgene, #52961). Sequence of sgRNAs is listed Supplementary Table 1.

**Antibodies.** All primary antibodies were used in TBST buffer with 5% non-fat milk for western blotting. Anti-AKT-pT308 antibody (13038, 1:1000), anti-AKT-pS473 antibody (4060, 1:3000), anti-AKT1 antibody (2938, 1:3000), anti-AKT pan antibody (4685, 1:3000), anti-pGSK-3β antibody (5558, 1:5000), anti-GSK-3β antibody (12456, 1:5000), anti-pFOXO antibody (2599, 1:1000), anti-PRMT1 (2449, 1:1000), anti-PRMT4 antibody (12495, 1:1000), anti-PRMT5 antibody (79998, 1:1000), anti-S6K1 antibody (9202, 1:2000), anti-E-cadherin antibody (3195, 1:3000), anti-PDK1 antibody (13037, 1:1000), anti-Ki-67 antibody (9027, 1:800 for IHC), anti-Myc-tag rabbit antibody (2278, 1:1000), anti-Glutathione S-transferase (GST) rabbit antibody (2625, 1:1000), anti-HA rabbit antibody (3724, 1:2000), and anti-Sdme-RG antibody (13222, 1:1000) were purchased from Cell Signaling Technology. Anti-FOXO3A antibody (10849-1-AP, 1:2000) and anti-Tubulin antibody (66240-1-Ig, 1:10,000) were purchased from Proteintech. Anti-sym11 antibody (07-413, 1:1000), anti-FLAG rabbit antibody (F7425, 1:2000), anti-FLAG mouse antibody (F3165, 1:2000), peroxidase-conjugated anti-mouse secondary antibody (A4416, 1:3000), and anti-rabbit secondary antibody (A4914, 1:3000) were obtained from Sigma. Monoclonal anti-HA antibody (901503, 1:3000) was purchased from Biolegend. Anti-PRMT2 antibody (720141, 1:500) was purchased from Thermo Fisher. Anti-PRMT3 antibody (ab191562, 1:10,000) was purchased from Abcam. Anti-PRMT6 antibody (sc-271744, 1:2000) was purchased from Santa Cruz Biotechnology. Anti-PRMT7 antibody (A12159, 1:2000) was purchased from ABclonal. Anti-PRMT9 antibody (A304-189A, 1:1000) was purchased from Bethyl Laboratories. Anti-AKT1-R391-me2s (1:1000) and anti-AKT1-R391-control antibody (1:1000) were generated by ABclonal.

**Immunoblot and immunoprecipitation analyses.** Cells were rinsed with ice-cold phosphate-buffered saline (PBS) and lysed with EBC buffer (50 mM Tris pH 7.5, 120 mM NaCl, 0.5% NP-40) or Triton buffer (40 mM HEPES pH 7.4, 150 mM NaCl, 2.5 mM MgCl₂, and 1% Triton) supplemented with protease inhibitor (Thermo Fisher, A32953) and phosphatase inhibitors (phosphatase inhibitor cocktail sets I and II, Calbiochem). The cell lysates were centrifuged at 13,200 r.p.m. for 10 min at 4 °C. The protein concentrations were measured using Nanodrop (Thermo Fisher) with Bio-Rad protein assay reagent. Equal amounts of whole cell lysates were resolved by SDS-polyacrylamide gel electrophoresis (SDS-PAGE) and immunoblotted with the indicated antibodies. For immunoprecipitation, the lysates (1000–3000 μg) were incubated with 50% slurry of agarose conjugated antibody for 3–5 h at 4 °C. The immunoprecipitates were washed four times with NETN buffer (20 mM Tris pH 8.0, 150 mM NaCl, 1 mM EDTA, and 0.5% NP-40) or Triton buffer before being resolved by SDS-PAGE. Anti-FLAG agarose beads (A2220) and anti-HA agarose beads (A2095) were purchased from Sigma. Anti-AKT1 antibody conjugated to agarose (sc-5298 AC) and anti-PRMT5 antibody conjugated to agarose (sc-376937 AC) were purchased from Santa Cruz Biotechnology.

**Purification of GST-tagged protein from bacteria.** Recombinant GST-GSK-3β protein was purified from the BL21(DE3) *Escherichia coli* transformed with pGEX-GSK-3β. Single colony was grown in 5 mL Luria-Bertani (LB) medium overnight at 37 °C. The culture was then 1:100 inoculated into 300 mL LB medium until an optical density of 0.4–0.6. The protein expression was induced by 0.1 mM IPTG (isopropyl-β-D-thiogalactoside) at 25 °C for 16 h. The bacteria cells were collected and re-suspended in GST buffer (25 mM Tris pH 8.0, 5 mM dithiothreitol (DTT), 150 mM NaCl) and sonicated (6 cycles of 5 s each at 50%). After centrifugation, the supernatant was incubated with glutathione sepharose beads for 3 h at 4 °C. The protein-bound glutathione beads were washed three times with GST buffer and eluted with elution buffer (10 mM L-Glutathione, 50 mM Tris-HCl pH 8.0).

**In vitro kinase assays.** AKT in vitro kinase assays were performed as previously described[64]. Briefly, 1 μg of bacterially purified GST-GSK-3β recombinant proteins were incubated with immunoprecipitated AKT1 from cells in kinase reaction buffer (1 μM MnCl₂, 2 mM DTT, 50 mM Tris pH 7.5) for 30 min at 30 °C. The reaction was stopped by adding 3× SDS loading buffer and was resolved by SDS-PAGE. Phosphorylation of GST-GSK-3β was detected by anti-pGSK-3β antibody (Cell Signaling Technology, 5558, 1:3000).

**In vitro methylation assays.** PRMT5 in vitro methylation assays were performed as previously described[66]. Briefly, 5 μg of recombinant GST-AKT1 proteins purified from HEK293T cells or GST-AKT1-KD-WT or R391K proteins purified from bacteria were incubated with immunoprecipitated HA-PRMT5 or HA-PRMT5-E444Q in the presence of 1 μL of adenosyl-L-methionine, S-[methyl-³H] (1 mCi/ml stock solution, Perkin Elmer). The reactions were performed in the methylation buffer (50 mM Tris pH 8.5, 20 mM KCl, 10 mM MgCl₂, 1 mM β-mercaptoethanol, and 100 mM sucrose) at 30 °C for 1 h and stopped by adding 3× SDS loading buffer and was resolved by SDS-PAGE. The separated samples were then transferred from the gel to a polyvinylidene difluoride membrane, which was then sprayed with EN³HANCE (Perkin Elmer) and exposed to X-ray film.

**Peptide synthesis and dot blot assays.** AKT1 peptides (386–395 aa) used for dot blot assays were synthesized at ABclonal. The sequences are as follows: R391-me0: KDPKQRLGGG; R391-me1:KDPKQ(R-me1)LGGG; R391-me2a:KDPKQ(R-me2a)LGGG; and R391-me2s:KDPKQ(R-me2s)LGGG. For dot blot assays, 1 μL diluted peptides were spotted onto a nitrocellulose membrane. The membrane was dried and were blocked with 5% non-fat milk for immunoblot analysis.

**Mass spectrometric analysis of AKT1-R391 methylation.** HEK293T cells were transfected with HA-AKT1. Forty-eight hours post transfection, the cells were lysed in Triton buffer, following HA immunoprecipitation. The immunoprecipitates were resolved by SDS-PAGE and visualized using GelCode blue staining reagent. The protein band containing HA-AKT1 was excised and digested with trypsin. Peptides were analyzed on an EASY nLC 1200 in-line with the Orbitrap Fusion Lumos Tribrid mass spectrometer (ThermoScientific). Peptides were pressure loaded at 800 bar and separated on a C18 reversed phase column (Acclaim PepMap RSLC, 75 μm × 50 cm (C18, 2 μm, 100 Å)) (Thermo Fisher) using a gradient of 2–35% B in 180 min (Solvent A: 0.1% FA; Solvent B: 80% ACN/0.1% FA) at a flow rate of 300 nL/min at 45 °C. Mass spectra were acquired in data-dependent mode with a high resolution (60,000) Fourier Transform mass spectrometry (FTMS) survey scan followed by MS/MS of the most intense precursors with a cycle time of 3 s. The automatic gain control target value was 4.0e5 for the survey MS scan. Fragmentation was performed with a precursor isolation window of 1.6 $m/z$, a maximum injection time of 50 ms, and higher-energy collisional dissociation (HCD) collision energy of 35%; the fragment ions were detected in the

Orbitrap at a 15,000 resolution. Spectra were searched against a custom database containing human AKT1 and a database of common contaminants using Max-Quant. The false discovery rate, determined using a reversed database strategy, was set at 1% at the peptide and modification site levels. Fully tryptic peptides with a minimum of seven residues were required including cleavage between lysine and proline. Two missed cleavages were permitted. Sites of modification were manually verified.

**IHC staining**. A breast cancer tissue microarray containing 100 cases of invasive carcinoma (BC081120f) was purchased from Biomax. The xenograft tumors were fixed in 10% neutral buffered formalin for 24 h and then paraffin-embedded and processed at the Histology & Immunohistochemistry Laboratory at Medical University of South Carolina (MUSC). Formalin-fixed, paraffin-embedded (FFPE) sections were deparaffinized using xylene and rehydrated in graded ethanol. Sections were heated in boiled citrate buffer (pH 6.0) for for 15 min. The remaining steps were performed using the ImmPRESS® Excel Amplified Polymer Kit (Vector Laboratories, MP-7601). Briefly, the sections were incubated with BLOXALL Blocking Solution (SP-6000) for 10 min, washed with wash buffer (10 mM sodium phosphate, pH 7.5, 0.9% saline and 0.1% Tween 20) for 5 min, and blocked with 2.5% Normal Horse Serum for 20 min. Sections were then incubated with anti-PRMT5 antibody (Proteintech, 18436-1-AP, 1:1000), anti-R391-me2s antibody (1:1000), or anti-Ki-67 antibody (Cell Signaling Technology, 9027, 1:800) diluted in normal horse serum for overnight at 4 °C. Sections were then washed with PBST buffer for 5 min and incubated with Goat anti-rabbit IgG (1:500) for 15 min at room temperature. After washing with PBST, sections were incubated with ImmPRESS Polymer Reagent for 30 min followed by washing with PBST. Afterward, all sections were developed using ImmPACT DAB EqV working solution until desired stain intensity and counterstained with hematoxylin, and mounted using SHURMount Mounting Media (General Data, #682188). The levels of PRMT5 and R391-me2s were determined by H-Score as described previously[67].

**TUNEL assay**. This assay was performed using the CF488A-TUNEL Apoptosis Assay Kit (Biotium, #30063). The FFPE sections were deparaffinized using xylene and rehydrated in graded ethanol. After washing with PBS, sections were permeabilized with 20 mg/mL proteinase K in PBS for 30 min at 37 °C. After incubating with 100 µL TUNEL Equilibration Buffer (Component 99965) for 5 min, sections were incubated with 50 µL of TUNEL reaction mix for 2 h at 37 °C and counterstained with 4′,6-diamidino-2-phenylindole (DAPI) and mounted using vibrance antifade mounting medium (Vector Laboratories, H-1700). The TUNEL-positive cells were determined by ImageJ.

**Immunofluorescence**. Cells grown on glass coverslips were fixed with 4% paraformaldehyde for 15 min at room temperature, washed three times with PBS, and then permeabilized with 0.05% Triton X-100 for 10 min at room temperature. Following three washes of 5 min in PBS, the coverslips were blocked with 5% bovine serum albumin for 30 min and then incubated with anti-AKT1 (Cell Signaling Technology, 2938, 1:300) or anti-AKT-pS473 (Cell Signaling Technology, 4060, 1:400) antibody for overnight at 4 °C. Following three washes of 5 min in PBST, the coverslips were incubated with goat anti-Rabbit IgG Alexa Fluor 488 (Invitrogen, #A-11008, 1:4 00) for 1 h at room temperature in the dark. Following three washes of 5 min with PBST, the coverslips were stained with DAPI and mounted using vibrance antifade mounting medium (Vector Laboratories, H-1700).

**Cell viability assays**. Two thousand cells/well were plated in 96-well plates for 24 h and then treated with GSK3326595 (MedChemExpress, HY-101563), MK2206 (Selleckchem, S1078), etoposide (Sigma, E1383), or cisplatin (Sigma, 232120) for 4 days. Fresh medium containing the drugs were replaced after 48 h. Survival cells were determined using the Cell Titer-Glo luminescent cell viability assay kit according to the manufacturer's instructions (Promega, G9242).

**Colony formation assays**. Cells were seeded in 6-well plates (400 cells per well) and cultured for 8–12 days until the formation of visible colonies. The colonies were fixed with 10% acetic acid and 10% methanol for 20 min, and then stained with 0.4% crystal violet in 20% ethanol for 20 min. After staining, the plates were washed with distilled water and air dried. The colonies were counted manually.

**Soft agar assays**. The anchorage-independent cell growth assay is a common method used to detect the ability of malignant transformation of cells and were performed as described previously[63]. Briefly, $1 \times 10^4$ or $2 \times 10^4$ cells were mixed with noble agar at a final concentration of 0.4% and layered over the bottom layer containing 0.8% noble agar. Completed DMEM medium (500 µl) was added to keep the top layer moist. The cells were then cultured for 3–4 weeks and stained with 1 mg/ml iodonitrotetrazolium chloride for colony visualization and counting manually.

**Mouse xenograft assays**. DLD-1-$AKT1/2^{-/-}$ ($2 \times 10^6$) cells stably expressing AKT1-WT or AKT1-R391K mutant or DLD-1 cells depleted PRMT5 and/or overexpressing myr-AKT1 were injected into the flank of 5-week-old female nude mice (The Jackson Laboratory). Tumor size was measured every 2 days with an electronic caliper. The tumor volume was calculated using the following formula: $L \times W^2 \times 0.5$, where $L$ is the longest diameter and $W$ is the shortest diameter. After 19–21 days, mice were killed. The solid tumors were dissected and weighed. All mice are housed in 22 °C, 50–60%, humidity and a 12 h-light/12 h-dark cycle. All mouse experiments were conducted under Protocol #IACUC-2018-00604 approved by the MUSC Institutional Animal Care and Use Committee.

**Statistical analysis**. The in vitro experiments were repeated at least two times unless stated otherwise. As indicated in the figure legends, all quantitative data are presented as the mean ± SD or mean ± SEM of three biologically independent experiments or samples. Statistical analyses were performed using GraphPad Prism 9 and Excel version 16.48. Statistical significance was determined by two-tailed Student's t-test or two-way analysis of variance. P-value < 0.05 was considered significant.

**Reporting summary**. Further information on research design is available in the Nature Research Reporting Summary linked to this article.

## Data availability

All relevant data are included in the paper and Supplementary Information files. Uncropped images for immunoblots are provided with this paper. The mass spectrometry raw data are available via ProteomeXchange with identifier PXD026071. Source data are provided with this paper.

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

## Acknowledgements

We thank members of the Dr. Long and Dr. Delaney Laboratory for helpful discussions. S.Y. is supported by Hollings Cancer Center Abney Postdoctoral Fellowship. S.K.O. is supported by CPRIT RR200030. This work was in part supported by pilot research funding from Hollings Cancer Center's Cancer Center Support Grant P30 CA138313 at the Medical University of South Carolina, and NIH grants S10 OD025126 (Orbitrap Fusion Lumos ETD MS), P20GM103542, and P30 DK123704 to L.E.B. and R00CA207867 to W.G.

## Author contributions

S.Y., L.L., and W.G. designed and performed the experiments with assistance from C.B. and V.P. L.E.B. performed the LC-MS/MS analysis of AKT1-R391 methylation. S.K.O. performed the structural modeling. M.C.O. and W.G. supervised the study. S.Y and W.G. wrote the manuscript with input from all authors.

## Competing interests

The authors declare no competing interests.
