## [Peer Review File · Nature Communications]

REVIEWER COMMENTS

Reviewer #1 (Remarks to the Author):

PRMT5 is the main symmetric di-methyltransferase in mammalian cells and has been increasingly linked to tumorigenesis. Recently, a number of papers have linked PRMT5 to the phosphorylation and activation of AKT (Zhang et al., *J Cell Mol Med*, 2019; Chung et al., *JBC*, 2019; Strobl et al., *Mol. Can. Therapeutics*, 2020; Li et al., *Cell Transplant.* 2019)). In this paper, Yin et al. investigate in more detail the link between PRMT5 and AKT and demonstrate that PRMT5 directly methylates AKT, that AKT methylation is induced by PI3K signalling, and that expression of methyl-deficient AKT suppresses *in vivo* tumour growth. They conclude that PRMT5 inhibitors can synergise with chemotherapies and AKT inhibitors promoting cancer cell death.

This is a well written paper; the data is of good quality and well controlled. Data presented in Figure 1 and 2 supports numerous other previous observations that PRMT5 regulates AKT phosphorylation and that this is functionally relevant, so although the data is nice, it is not particularly novel. Indeed, for me the most compelling finding is the mechanism by which arginine methylation promotes AKT phosphorylation through a shift from an AKT PH domain-in to PH domain-out conformation. This is an elegant demonstration of how methylation negatively regulates protein-protein interaction in the control of signalling pathways, and if this aspect of the paper had been developed further I would have been more supportive of publication. As it is, this paper is only incremental to the field, and, in its current form, may not be suitable for publication in *Nature Communications*.

Comments:

Is AKT2/3 phosphorylation also regulated by PRMT5 in MCF7 cells? Given the sequence alignment in Fig 3d, one would predict that PRMT5 might also be methylating AKT2 and AKT3. Is this the case?

The authors show that growth factors signalling increases methyl-AKT but do not explain how PRMT5 activity towards AKT is enhanced under such conditions.

Is AKT activation by oncogenic Ras PRMT5 dependent?

MCF7 cells have oncogenic mutations in PI3KCA. Is the regulation of AKT activity by PRMT5 apparent in cells expressing wild type PI3KCA? Are oncogenic PI3KCA mutated cancer cells more sensitive to PRMT5 inhibitors than wild type expressing cells?

The crystal structure of AKT1 demonstrating the inactive conformation has been solved (Wu et al., *Plos One* 2010). Can R-391me2s be modelled on such structure? Understanding the biochemical changes that occur between residues that are influenced by methylation altering number of H-bonds, pi-stacking or salt bridges would be very informative.

PRMT5 appears to be present within the membrane fraction (Fig 5b). Methods of how fractionation was performed are missing, so it is not clear if this is all cellular membranes or just the plasma membrane. If PRMT5 is constitutively found within the plasma membrane, do the authors have an explanation for this as generally PRMT5 is thought to be cytoplasmic/nuclear? Is this due to the fact that MCF7 cells most likely have constitutive AKT signalling because of activating PI3KCA mutations?

Inhibition of the PRMT5-AKT axis promotes apoptosis in other cell lines/tumour models. Is this also the case in their xenografts expressing AKT-R391K? Can more histological analysis of the tumours be presented?

The concentration of PRMT5 inhibitor used in Figure 7 is high (50uM) when the stated IC50 is 6.2nM. Moreover, addition of PRMT5 inhibitor alone only seems to have an effect in MCF7 cells (although statistical analysis is missing), and not MDA-MB-468, T47D and MDA-MB-231 cells. Consequently, I am not convinced that GSK3326595 is having much effect. The authors should repeat this analysis using GSK3203591 that has nM potency (Fong et al., *Cancer Cell*, 2019), and

link this to the presence/absence of PI3K/AKT activation mutations (as mentioned above).

A paper on BioRxiv has identified R15 within an RG motif of AKT1 as a PRMT5 methylation site in neuroblastoma cell lines (<https://www.biorxiv.org/content/10.1101/2020.08.12.246660v3>). Although the site of methylation may occur in a cell type-specific context, how convinced are the authors that R15 is not methylated by PRMT5 in their cells? Does methylation, as determined by in vivo labelling with 3H-methionine, decrease with the R15K mutant? Does expression of AKT-R15K in MCF7 cells affect cellular proliferation? Why are there these apparent discrepancies between which site is methylated by PRMT5?

Minor points:

Figure legend of Supp 6 I and J are omitted.

Reviewer #2 (Remarks to the Author):

In this study, Yin at al. reported a novel posttranslational modification of Akt1 termed Akt arginine methylation in Akt signaling activation and tumorigenesis. They showed that PRMT5 is a methyltransferase that elicits Akt1 methylation at R391, which is crucial for Akt1 kinase activation and oncogenic activity. The study is interesting and provides novel insight for oncogenic Akt signaling activation. However, several important questions remained to be addressed in order to validate the conclusions.

Major comments:

1. While PRMT5 methyltransferase activity is shown to be required for Akt methylation, there are no rescued experiments in PRMT knockout/knockdown cells demonstrating that PRMT5 enzymatic activity is required for Akt activation upon growth factor insulin and/or EGF treatment.
2. Is PI3K activity required for the interaction of Akt1 and PRMT5 interaction and subsequent Akt1 R391 methylation?
3. Do PRMT5 and Akt1R391 methylation regulate Akt1 binding to PIP3?
4. Does PRM5 directly methylate Akt1 at R391 in an in vitro methylation assay?
5. Akt membrane recruitment has been done in steady state condition. It is necessary to include growth factors like insulin or EGF in Fig. 5B and Fig. 5C.
6. Does loss of PRMT5 reduce tumorigenesis formation? If so, does Akt regulation by PRMT5 is involved in this process. Xenograft assays for PRMT5 loss and Mri-Akt1 rescued experiments should be included.
7. The authors should provide more detailed insight into how Akt1 R391 methylation regulates Akt intramolecular interaction.
8. The clinical relevance of this study has not been well articulated. Does PRMT5 overexpression correlate with Akt1 R391 methylation in cancer specimens?
9. Since PRMT5 targeting impair Akt activity/activation, what is the rationale to combine Akt inhibitor with PRMT5 inhibitor cancer targeting? The conclusion for the synergistic effect claimed in Fig. 7a,7b, 7f, 7i and supp 7d is not well supported by the data. It appears to be additive effect instead of synergistic effect.

Minor points.

1. PRMT5 expression in Fig. 2c should be shown.
2. Akt pT308 and pS473 blots are poor. The R391-Me2s blot for Fig. 4C and PDK1 blot for 5f in IP samples are also poor.

Reviewer #3 (Remarks to the Author):

I have read the manuscript entitled as "PRMT5-mediated arginine methylation activates AKT kinase to govern tumorigenesis" by Drs Wenjian Gan and coworkers.

In this manuscript, authors show that protein arginine methyltransferase 5 (PRMT5) is a methyltransferase of AKT

1 PRMT5 inhibition (Crisper or shRNA) suppresses AKT activation (Fig 1).

2 AKT rescues cell proliferation in PRMT5-depleted cells (Fig 2).

3 PRMT5 catalyzes symmetric dimethylation of AKT1-R391 (Fig 3).

4 Active form of AKT enhances R391 dimethylation (Fig 4).

5 Absence of R391 methylation blocks AKT activation processes (Fig 5).

6 Absence of R391 methylation attenuates AKT activation and tumorigenesis (fig 6).

7 PRMT5 inhibition sensitizes breast cancer cells to AKT inhibitor and chemotherapeutic agents (Fig 7).

Thus authors conclude that R391 methylation is an important step for AKT activation and its oncogenic function.

In general, studies are logically designed and carefully performed and technically sound, which supported the conclusion.

There are few suggestions may need attention.

It would be better to add size markers on the side of the immunoblot all the way across the manuscript.

AKT is activated by PDK1 on T 308, and mTORC2 on S-473. In the manuscript, it is well documented that PRMT5 can enhance the PDK1 dependent AKT phosphorylation. How mechanistically PRMT5 can enhance the mTORC2 mediated AKT phosphorylation at S-473 (independent from PDK1) may not be well demonstrated. The analysis using TSC1/TSC2 null cells may help to dissect that direction, perhaps.

Fig 4d. The expression of AKT WT is away higher than R25C (second panel, middle lane), which may enhance the interaction between PRMT5 (top panel, middle lane).

In addition to the biochemical data to localize the methylation dependent localization of p-AKT authors are encouraged to show by immunofluorescent study for the intracellular localization of p-AKT will be preferable.

Can authors show the interaction between PIP3 with WT-AKT vs R391K-AKT by PIP strip, for example.

Fig. 5 Does PDK1 directly interact with PRMT5 or through PDK1-PIP3 interaction?

Fig 3 legend lane 2 c. In vitro "I" seemed to be bold, perhaps.

Fig 6b. Third panel from the top. WT AKT expression appear to be slightly more than R391K. Can authors confirm expression is even by HA blot, which is tagged to AKT?

Point-by-Point Response to the Reviewers' comments
(Manuscript ID: NCOMMS-20-41840-T)

We sincerely appreciate the thorough analyses and constructive comments provided by three reviewers, which have helped us further improve our manuscript. As described in detail below, we have obtained substantial new experimental evidence to fully address all the reviewers' concerns. We hope the editor and the reviewers will concur with us that we have satisfactorily addressed all the raised concerns and consequently, substantially strengthened our paper for publication.

Reviewer #1 (Remarks to the Author):

PRMT5 is the main symmetric di-methyltransferase in mammalian cells and has been increasingly linked to tumorigenesis. Recently, a number of papers have linked PRMT5 to the phosphorylation and activation of AKT (Zhang et al., J Cell Mol Med, 2019; Chung et al., JBC, 2019; Strobl et al., Mol. Can. Therapeutics, 2020; Li et al., Cell Transplant. 2019)). In this paper, Yin et al. investigate in more detail the link between PRMT5 and AKT and demonstrate that PRMT5 directly methylates AKT, that AKT methylation is induced by PI3K signalling, and that expression of methyl-deficient AKT suppresses in vivo tumour growth. They conclude that PRMT5 inhibitors can synergise with chemotherapies and AKT inhibitors promoting cancer cell death.

This is a well written paper; the data is of good quality and well controlled. Data presented in Figure 1 and 2 supports numerous other previous observations that PRMT5 regulates AKT phosphorylation and that this is functionally relevant, so although the data is nice, it is not particularly novel. Indeed, for me the most compelling finding is the mechanism by which arginine methylation promotes AKT phosphorylation through a shift from an AKT PH domain-in to PH domain-out conformation. This is an elegant demonstration of how methylation negatively regulates protein-protein interaction in the control of signalling pathways, and if this aspect of the paper had been developed further I would have been more supportive of publication. As it is, this paper is only incremental to the field, and, in its current form, may not be suitable for publication in Nature Communications.

Response: We thank the reviewer for recognizing the novelty and the convincing data of our study that provides the molecular mechanism of PRMT5-mediated AKT activation and tumorigenesis. We also thank the reviewer for the excellent suggestions to guide us to further improve our study.

We fully agree with the reviewer that several previous studies have reported that PRMT5 is linked with AKT signaling. Both studies (Zhang et al., J Cell Mol Med, 2019 and Li et al., Cell Transplant. 2019) showed that down-regulation PRMT5 by shRNA or PRMT5 inhibitor reduced AKT phosphorylation and cell proliferation in lung cancer cells. Chung et al., JBC, 2019 showed that knockdown of *PRMT5* by shRNA decreased AKT phosphorylation. Strobl et al., Mol. Can. Therapeutics, 2020 reported that PRMT5 inhibitor impaired the AKT pathway. However, as the reviewer kindly mentioned, these studies did not provide a molecular mechanism underlying how PRMT5 regulates AKT activation. Moreover, these studies also did not investigate whether AKT

is critical for PRMT5-mediated cell proliferation and tumor growth. Therefore, our study significantly advanced the current understanding of the PRMT5/AKT axis in the following aspects:

- 1) We identified AKT as a critical downstream effector that mediates the oncogenic function of PRMT5. The newly obtained results in *revised Fig. 2e-f* showed that knockdown of *PRMT5* significantly suppresses tumor growth, which can be rescued by ectopic expression of the constitutively active myr-AKT1.
- 2) We characterized PRMT5 as a physiological methyltransferase to directly methylate AKT1 at R391.
- 3) We provided a novel regulatory mechanism that AKT1-R391 methylation facilitates AKT conformational shift from PH domain-in to PH domain-out state and subsequent AKT membrane recruitment and phosphorylation by PDK1 and mTORC2.

We also fully agree with the reviewer that additional experimental evidence is necessary to strengthen the notion that AKT1-R391 methylation facilitates a shift from PH domain-in to PH domain-out confirmation. To this end, we have obtained the following new experimental results:

- 1) Knockdown of *PRMT5* enhanced PH-KD interaction (*revised Supplementary Fig. 5a*). On the other hand, ectopic expression of PRMT5/MEP50 reduced PH interaction with KD-WT, but not KD-R391K (*revised Supplementary Fig. 5b*).
- 2) Interestingly, PRMT5 perturbed KD interaction with PH-R25C mutant (*Supplementary Fig. 5c*), indicating that PRMT5 promotes AKT confirmation change independent of its binding to PIP3.
- 3) Importantly, peptides without R391 modification displayed stronger interaction with PH domain than peptides containing R391 symmetric dimethylation (*revised Fig. 5b*), providing direct evidence that R391 symmetric dimethylation disrupts its binding to PH.
- 4) Moreover, R391K mutation or depletion of *PRMT5* attenuated AKT1 interaction with PIP3 (*revised Fig. 5c, d*), further supporting that deficiency in AKT1-R391 methylation locks AKT in the PH-in state.

Taken together, we hope the reviewer would agree with us that we have provided additional experimental evidence to strengthen the conceptual advances we have made in our current study.

Comments:

1. Is AKT2/3 phosphorylation also regulated by PRMT5 in MCF7 cells? Given the sequence alignment in Fig 3d, one would predict that PRMT5 might also be methylating AKT2 and AKT3. Is this the case?

Response: We thank the reviewer for raising these excellent questions and fully agree with the reviewer that PRMT5 might also promote AKT2/3 methylation and phosphorylation. To support this notion, we have obtained new experimental data listed below:

- 1) As suggested, we intended to examine whether PRMT5 regulates AKT2/3 phosphorylation in MCF7 cells. However, we found that only AKT1/2, but not AKT3, could be detected in MCF7 cells (*revised Supplementary Fig. 1m*). Fortunately, we could detect AKT1/2/3 signal in MDA-MB-231 breast cancer cells (*revised Supplementary Fig. 1m*). Therefore, MDA-MB-231 instead of MCF7 cells were used to address the question raised by the reviewer. We

generated *AKT1/2-KO* and *AKT1/3-KO* cells to retain only AKT3 and AKT2, respectively (*revised Supplementary Fig. 1n*). Notably, compared to control, knockout of *PRMT5* in these cells decreased AKT phosphorylation at both T308 and S473 (*revised Supplementary Fig. 1o, p*), suggesting that *PRMT5* also regulates AKT2/3 phosphorylation in MDA-MB-231 cells.

- 2) Consistent with the finding that all AKT isoforms bind to *PRMT5* (*revised Supplementary Fig. 3b*), AKT2/3 are also methylated in cells, which can be blocked by mutating the corresponding residue AKT2-R392K and AKT3-R388K (*revised Supplementary Fig. 3o, p*).
- 3) As a result, we showed that AKT2-R392K and AKT3-R388K mutation suppress AKT phosphorylation at both T308 and S473 (*revised Supplementary Fig. 3o, p*), further supporting that like AKT1, *PRMT5*-mediated methylation of AKT2/3 is also important for their activation.

These results demonstrate that *PRMT5* interacts with all the AKT isoforms and promotes their arginine methylation and activation.

2. The authors show that growth factors signaling increases methyl-AKT but do not explain how *PRMT5* activity towards AKT is enhanced under such conditions.

Response: We thank the reviewer for raising this concern. Our data listed below support the model that growth factors stimulate PIP3-dependent release of KD from PH, in turn, promote *PRMT5* access to AKT-KD for R391 methylation.

- 1) Growth factors including insulin and EGF enhance interaction between *PRMT5* and AKT (*revised Fig 4a and Supplementary Fig. 4a, b*).
- 2) PIP3 enhances *PRMT5* interaction with AKT1-WT, but not the non-PIP3 binding mutant, AKT1-R25C in vitro (*revised Fig. 4f*).
- 3) Consistently, AKT-R25C mutation impairs AKT1 interaction with *PRMT5* (*revised Fig. 4d*).
- 4) On the other hand, AKT1-E17K mutation, which has a higher affinity of PIP2/3, enhances AKT1 binding to *PRMT5* (*revised Supplementary Fig. 4f*).
- 5) Furthermore, growth factors have minimal effect on H4R3-me2s (*PRMT5* substrate), indicating growth factors may not affect *PRMT5* methyltransferase activity (*revised Supplementary Fig. 4d, e*).

Together, these results indicate that growth factors promote AKT1-R391 methylation primarily by enhancing *PRMT5* interaction with AKT due to the conformation change (from PH-in to PH-out).

3. Is AKT activation by oncogenic Ras *PRMT5* dependent?

Response: We thank the reviewer for raising this question. The study from Dr. Khavari group (Kovalski et al., Mol Cell. 2019) found that the oncogenic Ras mutant directly associates with mTOR and MAPKAP1 (also called Sin1, a specific subunit of mTORC2 complex) to promote mTORC2 kinase activity, leading to an increase of AKT-pS473. Following this direction, we have generated MCF7 cells stably expressing NRAS-WT or the oncogenic mutant NRAS-G12V. Notably, knockout of *PRMT5* in these cells reduced the phosphorylation of AKT-S473 (*revised Supplementary Fig. 7h*), which may be due to the reduced binding between AKT1 and Sin1 in

PRMT5-depleted cells (*revised Supplementary Fig. 6f*). These results suggest that *PRMT5* also plays a critical role in oncogenic Ras-mediated AKT activation.

4. MCF7 cells have oncogenic mutations in *PI3KCA*. Is the regulation of AKT activity by *PRMT5* apparent in cells expressing wild type *PI3KCA*? Are oncogenic *PI3KCA* mutated cancer cells more sensitive to *PRMT5* inhibitors than wild type expressing cells?

Response: We thank the reviewer for raising these important questions and fully agree with the reviewer that addressing these questions will further improve our study. To this end, we observed that knockout of *PRMT5* has similar inhibition of AKT phosphorylation in cells harboring *PI3KCA* mutation [MCF7 (E545K) and T-47D (H1047R)] and cells harboring *PI3KCA* wildtype (MDA-MB-231 and MDA-MB-436) (*revised Fig. 1a and Supplementary Fig. 1h-j*). Moreover, T-47D (*PI3KCA* mutation), BT-549 (*PI3KCA* WT) and MDA-MB-231 (*PI3KCA* WT) cells, or MCF7 (*PI3KCA* mutation) and MDA-MB-468 (*PI3KCA* WT) cells displayed similar sensitivity to *PRMT5* inhibitor (*revised Fig. 7*). These results demonstrate that there is no apparent association between *PI3KCA* mutation and *PRMT5*-regulated AKT activity or sensitivity to *PRMT5* inhibitor in these breast cancer cells we examined. More studies using more breast cancer cell lines and mouse models will help also explain whether *PI3KCA* mutations dictate the efficacy of *PRMT5* inhibitors.

5. The crystal structure of AKT1 demonstrating the inactive conformation has been solved (Wu et al., Plos One 2010). Can R-391me2s be modelled on such structure? Understanding the biochemical changes that occur between residues that are influenced by methylation altering number of H-bonds, pi-stacking or salt bridges would be very informative.

Response: We fully agree with the reviewer that understanding biochemical changes caused by R391-me2s will be very informative. To this end, we analyzed AKT1 structures showing that R391 participates in a conserved network of salt bridges and hydrogen bonds with E319, A329, and K386 (*revised Supplementary Fig. 6h*). Notably, this serves to position a loop that harbors the conserved 'DYG' motif, which participates in a network of interactions with the AKT1 activation loop and the T308 side chain (PDB: 3CQW) (Lippa et al., Bioorg Med Chem Lett. 2008) and serves as a crucial regulatory region of many kinases (Treiber et al., Chem Biol. 2013). Modeling of symmetrically dimethylated Arg391 onto AKT1 shows that this network of interactions is disrupted both due to a loss of hydrogen bond donors on the dimethylated R391 side chain as well as steric clashes that occur (*revised Supplementary Fig. 6i*). Given the connection of this region to the DYG motif and activation loop, which includes T308, we speculate that in addition to the effect on PH/KD interaction, dimethylation of R391 may also promote T308 phosphorylation via an allosteric effect. A follow-up structural study in the future will further confirm our speculation.

6. *PRMT5* appears to be present within the membrane fraction (Fig 5b). Methods of how fractionation was performed are missing, so it is not clear if this is all cellular membranes or just the plasma membrane. If *PRMT5* is constitutively found within the plasma membrane, do the authors have an explanation for this as generally *PRMT5* is thought to be

cytoplasmic/nuclear? Is this due to the fact that MCF7 cells most likely have constitutive AKT signaling because of activating PI3KCA mutations?

Response: We apologize for the missing of detail for the cell fractionation assay, which has been included in the Methods section in the revised manuscript. Cell fractionation was performed using Subcellular Protein Fractionation Kit following the manufacturer's instructions (Thermo Fisher, 78840). The membrane fraction contains plasma, mitochondria and ER/Golgi membranes, but not nuclear membranes.

We agree with the reviewer that PRMT5 generally localizes in cytoplasmic and nuclear, and its subcellular localization is cell type-specific (Ancelin et al., Nat Cell Biol. 2006; Herrmann et al., J Cell Sci. 2009; Gu et al., PLoS One. 2012; Koh et al., Epigenetics. 2015). Interestingly, Zhou et al. reported that PRMT5 localizes to the Golgi apparatus in multiple cell types including U2OS, MCF7, HeLa, and HCT116. They found that PRMT5 is present in the Golgi membranes (Zhou et al., Cell Res. 2010). Therefore, the PRMT5 signal in the membrane fraction in our study is probably derived from Golgi membranes.

As suggested by the reviewer, we also examine whether the membrane localization of PRMT5 is associated with *PI3KCA* mutations. To this end, we performed the cell fractionation assay in breast cancer cells harboring *PI3KCA*-WT (MDA-MB-468 and HCC1937) and *PI3KCA*-mutation (T-47D and MCF7). We found that T-47D and HCC1937 cells as a group, while MCF7 and MDA-MB-468 cells as another group display similar levels of membrane-associated PRMT5 (**Fig. R1**). These results suggest that there is no apparent association between *PI3KCA* mutation and PRMT5 membrane localization in these breast cancer cells we examined. We agree with the reviewer that it will be interesting to investigate how upstream signals regulate PRMT5 membrane localization. However, we hope the reviewer will agree with us that this is not likely a major focus of this study, therefore warranting a follow-up study in a separate manuscript in the future.

Fig. R1. IB analysis of cell fractions derived from *PI3KCA*-WT and *PI3KCA*-mutation breast cancer cells.

7. Inhibition of the PRMT5-AKT axis promotes apoptosis in other cell lines/tumour models. Is this also the case in their xenografts expressing AKT-R391K? Can more histological analysis of the tumours be presented?

Response: We thank the reviewer for raising this critical question and for providing insightful suggestions. We fully agree with the reviewer that both PRMT5 and AKT are critical regulators of apoptosis. As suggested, we have performed TUNEL assay (Kyrylkova et al., Methods Mol Biol. 2012) to examine cell apoptosis in tumors derived from xenograft mouse assay. We have also assessed cell proliferation in these tumors by examining the expression of Ki-67 using immunohistochemistry staining, a well-acknowledged marker for proliferation (Scholzen and

Gerdes, J Cell Physiol. 2000). Notably, tumors expressing AKT-R391K contain more apoptotic cells and less proliferative cells than tumors expressing AKT-WT (*revised Fig. 6j-l*).

8. The concentration of PRMT5 inhibitor used in Figure 7 is high (50uM) when the stated IC50 is 6.2nM. Moreover, addition of PRMT5 inhibitor alone only seems to have an effect in MCF7 cells (although statistical analysis is missing), and not MDA-MB-468, T47D and MDA-MB-231 cells. Consequently, I am not convinced that GSK3326595 is having much effect. The authors should repeat this analysis using GSK3203591 that has nM potency (Fong et al., Cancer Cell, 2019), and link this to the presence/absence of PI3K/AKT activation mutations (as mentioned above).

Response: We thank the reviewer for raising this important concern and for providing insightful suggestions. We fully agree with the reviewer that the concentration of PRMT5 inhibitor we used was too high, and that the anti-proliferative effect of PRMT5 inhibitor GSK3326595 is not as expected in MDA-MB-468, T47D and MDA-MB-231 cells under the condition we used. After carefully reading the paper (Fong et al., Cancer Cell, 2019), we found that they treated the cells with PRMT5 inhibitor for 6 days, while we treated the cells for 2 days. To determine whether the difference in the treatment time is the major cause of the lower efficiency of GSK3326595 in our cells, we have repeated the cell viability assays by treating cells with different doses of GSK3326595 (0, 0.5, 1, 2.5 and 5 μ M) for 2, 4 or 6 days. Notably, the extension of treatment time rather than the increase of concentration significantly decreased cell survival in all cells we examined (*revised Supplementary Fig. 9a-d*), consistent with previous study by Dr. Olena Barbash group at GSK (Gerhart et al., Sci Rep. 2018). Moreover, we did not observe a dramatic difference on PRMT5 inhibitor sensitivity in cells harboring *PI3KCA*-WT (MDA-MB-231 and BT-549) and *PI3KCA*-mutation (T-47D), although MCF7 cells were more sensitive to PRMT5 inhibitor than other cells. Collectively, these newly obtained results demonstrated that the treatment time is a key factor determining the effect of the PRMT5 inhibitor.

We have also repeated the Fig. 7 using the newly identified conditions (treatment of 0.5 μ M GSK3326595 for 4 days) and found that GSK3326595 sensitized breast cancer cells to AKT inhibitor, etoposide and cisplatin (*revised Fig. 7*).

9. A paper on BioRxiv has identified R15 within an RG motif of AKT1 as a PRMT5 methylation site in neuroblastoma cell lines (<https://www.biorxiv.org/content/10.1101/2020.08.12.246660v3>). Although the site of methylation may occur in a cell type-specific context, how convinced are the authors that R15 is not methylated by PRMT5 in their cells? Does methylation, as determined by in vivo labelling with 3H-methionine, decrease with the R15K mutant? Does expression of AKT-R15K in MCF7 cells affect cellular proliferation? Why are there these apparent discrepancies between which site is methylated by PRMT5?

Response: We thank the reviewer for pointing out these important discrepancies between our study and the paper on BioRxiv. In this paper, the authors showed that R15 is methylated in neuroblastoma cell lines as detected by the anti-SDMA antibody. However, as the authors did not provide the anti-SDMA antibody information in their paper, we are not sure whether they used the

same antibody as the one we used. Moreover, the authors also mentioned that there is a weak SDMA signal in R15K mutant in neuroblastoma cells, indicating that there is other arginine methylation site(s) in AKT1. Our study found that AKT1-WT and AKT1-R15K displayed similar methylation signals in MCF7 cells as detected by the sym11 antibody that was widely used to examine symmetric arginine dimethylation (*revised Supplementary Fig. 3j*). We have also examined the methylation of AKT-R15K in MCF7 cells using another anti-Symmetric Di-Methyl Arginine Motif [Sdme-RG] antibody purchased from Cell Signaling Technology and confirm that R15K mutation did not reduce Sdme-RG signal (*revised Supplementary Fig. 3k*).

We also agree with the reviewer that it is possible to determine AKT methylation using the ³H-methionine labeling in vivo methylation assay in cells. However, this method can detect both lysine and arginine methylation, and AKT has been reported to be methylated at multiple lysine by SETDB1 (Guo et al., Nat Cell Biol. 2019; Wang et al., Nat Cell Biol. 2019). Therefore, AKT1 lysine methylation may interfere with the analysis of AKT-R15 methylation in cells by in vivo methylation assay. In this case, we fully agree with the reviewer to determine whether the expression of AKT-R15K in MCF7 cells affects cellular proliferation. To this end, we re-expressed AKT1-WT, R15K or R391K in MCF7-*AKT1/2*-KO cells. Notably, AKT-R15K mutant displayed comparable phosphorylation levels at T308 and S473 as AKT1-WT (*revised Supplementary Fig. 8i*). Moreover, R15K mutation did not significantly affect cell proliferation and colony formation, compared to AKT1-WT (*revised Supplementary Fig. 8j, k*). These results suggest that methylation of AKT1-R391, but not R15, is critical for AKT activation and cell proliferation in MCF7 cells.

Altogether, we fully agree with the reviewer that the site of methylation in AKT1 may occur in a cell type-specific context. To eliminate possible confusion for the readers, we have discussed this possibility in the revised manuscript.

Minor points:

Figure legend of Supp 6 I and J are omitted.

Response: We thank the reviewer for picking up this mistake and apologize for missing Figure legend of Supp 6i and j. We have included the corresponding figure legends in the revised Supplementary Fig. 8l, m and carefully checked all other figure legends to avoid such mistakes.

Reviewer #2 (Remarks to the Author):

In this study, Yin et al. reported a novel posttranslational modification of Akt1 termed Akt arginine methylation in Akt signaling activation and tumorigenesis. They showed that PRMT5 is a methyltransferase that elicits Akt1 methylation at R391, which is crucial for Akt1 kinase activation and oncogenic activity. The study is interesting and provides novel insight for oncogenic Akt signaling activation. However, several important questions remained to be addressed in order to validate the conclusions.

Response: We thank the reviewer for recognizing the novelty and significance of our study that AKT1-R391 methylation is a novel regulatory mechanism of AKT activation and its oncogenic function. We also thank the reviewer for the excellent suggestions to guide us to further improve our study.

Major comments:

1. While PRMT5 methyltransferase activity is shown to be required for Akt methylation, there are no rescued experiments in PRMT knockout/knockdown cells demonstrating that PRMT5 enzymatic activity is required for Akt activation upon growth factor insulin and/or EGF treatment.

Response: We thank the reviewer for raising this important concern and fully agree with the reviewer that the rescued experiment is important to support our conclusion. As suggested, we reintroduced PRMT5-WT or the PRMT5-E444Q mutant (the enzymatically dead form of PRMT5) into *PRMT5*-depleted MCF7 cells. Notably, PRMT5-WT, but not the PRMT5-E444Q mutant, could rescue AKT activation in *PRMT5*-depleted cells at normal cell culture condition or upon insulin treatment (*revised Supplementary Fig. 2c, d*), as well as cell colony formation (*revised Supplementary Fig. 2e*). These results demonstrate that PRMT5 enzymatic activity is required for AKT activation and cell proliferation.

2. Is PI3K activity required for the interaction of Akt1 and PRMT5 and subsequent Akt1 R391 methylation?

Response: We thank the reviewer for raising this important question. As suggested, we found that insulin or EGF treatment enhances PRMT5 interaction with AKT1 and consequently promotes AKT1-R391 methylation (*revised Fig. 4a, b and Supplementary Fig. 4a-c*), which could be reversed by PI3K inhibitors including LY294002, Wortmannin and Buparlisib (*revised Fig. 4c*). Moreover, PIP3 enhances PRMT5 interaction with AKT1-WT, but not the AKT1-R25C mutant in vitro (*revised Fig. 4f*). This result suggests that PI3K signaling may enhance PRMT5-mediated AKT1-R391 methylation through PIP3-induced AKT conformational change.

3. Do PRMT5 and Akt1R391 methylation regulate Akt1 binding to PIP3?

Response: We thank the reviewer for raising this question. As suggested, we have obtained new experimental data showing that PRMT5-mediated AKT1-R391 methylation promotes AKT1 binding to PIP3. Specifically, we used the P(3,4,5) PIP beads (Echelon Biosciences, #P-B345A) to pulldown AKT1 as described previously (Guo et al., Nat Cell Biol. 2019). Notably, R391K mutation or depletion of *PRMT5* dramatically reduced AKT1 binding to PIP3 (*revised Fig. 5c, d*). These data indicate that PRMT5-mediated R391 methylation may facilitate AKT1 binding to PIP3 to induce AKT conformational shift from the PH-in to PH-out state.

4. Does PRMT5 directly methylate Akt1 at R391 in an in vitro methylation assay?

Response: We thank the reviewer for raising this critical question and fully agree with the reviewer that it will further strengthen our study by adding such data. To this end, we tried to purify the full length of AKT1 protein from *E. coli* instead of mammalian cells to avoid interference from AKT-binding proteins or posttranslational modifications in mammalian cells. However, we could not obtain the full length of AKT1 protein from *E. coli* even we tried many conditions. Alternatively, as PRMT5 mainly binds to the AKT1-KD where R391 is located, we purified the GST-AKT1-KD proteins from *E. coli* for the *in vitro* methylation assay. Consistent with R391K mutation blocks PRMT5-mediated methylation in cells (*revised Fig. 3f*), AKT1-KD proteins containing R391K mutation fail to be methylated by PRMT5 *in vitro* (*revised Fig. 3e*). These results demonstrate that R391 is the major methylation site of AKT1 by PRMT5.

5. Akt membrane recruitment has been done in steady state condition. It is necessary to include growth factors like insulin or EGF in Fig. 5B and Fig. 5C.

Response: We thank the reviewer for this constructive suggestion. To this end, we have performed the experiments in Fig. 5b and Fig. 5c with insulin treatment. Notably, AKT1 membrane translocation was increased by insulin stimulation in sgGFP cells and AKT1-WT expressing cells, but not *PRMT5*-depleted cells and AKT1-R391K expressing cells (*revised Supplementary Fig. 5e, f*). We have also performed an immunofluorescent study to confirm that R391K mutation decreases AKT1 membrane localization upon insulin stimulation (*revised Supplementary Fig. 5d*).

6. Does loss of PRMT5 reduce tumorigenesis formation? If so, does Akt regulation by PRMT5 is involved in this process. Xenograft assays for PRMT5 loss and Myr-Ak1 rescued experiments should be included.

Response: We thank the reviewer for raising these concerns and for providing excellent suggestion. As suggested, we have performed xenograft mouse assay by injecting the cells into the flank of nude mice, including EV+shGFP, EV+shPRMT5, myr-AKT1+shGFP and myr-AKT1+shPRMT5. Consistent with the results of cell colony formation assay and soft agar assay (*revised Fig. 2b, c*), knockdown of *PRMT5* significantly retarded tumor growth, compared to the control (shGFP). Importantly, ectopic expression of myr-AKT1 in *PRMT5*-depleted cells restored tumor growth (*revised Fig. 2d, e*). Moreover, we performed TUNEL assay and immunohistochemistry (IHC) staining of Ki-67 in these tumors. Notably, tumors with PRMT5 depletion contain more apoptotic

cells and less Ki-67 positive cells than control tumors (shGFP), which could be rescued by expressing myr-AKT1 (*revised Fig. 2f-h*). These results suggest that AKT is a major downstream effector mediating the oncogenic function of PRMT5.

7. The authors should provide more detailed insight into how Akt1 R391 methylation regulates Akt intramolecular interaction.

Response: We fully agree with the reviewer that additional experimental evidence is necessary to support the notion that AKT1-R391 methylation regulates AKT intramolecular interaction. To this end, we have obtained the following new experimental results:

- 1) Knockdown of *PRMT5* enhanced PH-KD interaction (*revised Supplementary Fig. 5a*). On the other hand, ectopic expression of PRMT5/MEP50 reduced PH interaction with KD-WT, but not the KD-R391K mutant (*revised Supplementary Fig. 5b*).
- 2) Interestingly, PRMT5 perturbed KD interaction with PH-R25C mutant (*Supplementary Fig. 5c*), indicating that PRMT5 promotes AKT confirmation change independent of its binding to PIP3.
- 3) Importantly, peptides without R391 modification displayed stronger interaction with PH domain than peptides containing R391 symmetric dimethylation (*revised Fig. 5b*), providing direct evidence that R391 symmetric dimethylation disrupts KD binding to PH.
- 4) Moreover, R391K mutation or depletion of *PRMT5* attenuated AKT1 interaction with PIP3 (*revised Fig. 5c, d*), further supporting that deficiency in AKT1-R391 methylation locks AKT in the PH-in state.

Taken together, we hope the reviewer would agree with us that we have provided additional experimental evidence to strengthen the conceptual advances we have made in our current study.

8. The clinical relevance of this study has not been well articulated. Does PRMT5 overexpression correlate with Akt1 R391 methylation in cancer specimens?

Response: We thank the reviewer for raising this concern and for providing instructive suggestions. To this end, we have performed IHC staining to examine PRMT5 expression levels and AKT1-R391 methylation in breast cancer patient specimens. Notably, higher expression levels of PRMT5 are positively correlated with higher levels of AKT1-R391-me₂s (*revised Fig. 3h, i*), further supporting that PRMT5 is a methyltransferase of AKT1-R391.

9. Since PRMT5 targeting impair Akt activity/activation, what is the rationale to combine Akt inhibitor with PRMT5 inhibitor cancer targeting? The conclusion for the synergistic effect claimed in Fig. 7a,7b, 7f, 7i and supp 7d is not well supported by the data. It appears to be additive effect instead of synergistic effect.

Response: We thank the reviewer for raising these concerns. Although many AKT inhibitors including AZD5363, MK2206, and GDC-0068 are in phase I/II clinical trials, especially for breast cancers, none of these inhibitors has achieved satisfactory outcomes in monotherapy in part due to

the inadequate AKT inhibition at tolerable doses. Therefore, the combination of AKT inhibitors with chemotherapy and other targeted therapy is a potential option for cancer treatment (Nitulescu et al., Int. J. Oncol. 2016). Given PRMT5 is one of critical AKT regulators, we tested the synergistic effect of AKT inhibitor and PRMT5 inhibitor. We have included these discussions in the revised manuscript for the rationale to combine AKT inhibitor with PRMT5 inhibitor.

We agree with the reviewer that the results in original Fig. 7a,7b, 7f, 7i and supp 7d are not convincing. As mentioned by Reviewer #1, the concentration of PRMT5 inhibitor we used is too high and that the anti-proliferative effect of PRMT5 inhibitor GSK3326595 is not as expected in MDA-MB-468, T47D and MDA-MB-231 cells in our assays. Following the suggestion by reviewer #1, we have carefully read the paper that tests the efficiency of PRMT5 inhibitors in multiple cell lines (Fong et al., Cancer Cell, 2019) and found that they treated the cells with PRMT5 inhibitor for 6 days, while we treated the cells for 2 days. To determine whether the difference in the treatment time is a major cause of the lower efficiency of GSK3326595 in our cell viability assays, we have repeated these assays by treating cells with different doses of GSK3326595 (0, 0.5, 1, 2.5 and 5 μ M) for 2, 4 or 6 days. Notably, the extension of treatment time rather than the increase of concentration significantly decreased cell survival in all cells we examined (*revised Supplementary Fig. 9a-d*), consistent with previous study by Dr. Olena Barbash group at GSK (Gerhart et al., Sci Rep. 2018). Thus, we have repeated the assays in Fig. 7 by treating cells with 0.5 μ M GSK3326595 and/or other drugs for 4 days. Consistently, PRMT5 inhibitor sensitized cells to MK2206 (AKT inhibitor), etoposide and cisplatin in breast cancer cells we examined (*revised Fig. 7a-f*).

Minor points.

1. PRMT5 expression in Fig. 2c should be shown.

Response: We thank the reviewer for providing this suggestion. We have examined the expression of PRMT5 as well as AKT phosphorylation (*revised Supplementary Fig. 2c, d*) in the cells in the original Fig. 2c, which is now revised Supplementary Fig. 2e.

2. Akt pT308 and pS473 blots are poor. The R391-Me2s blot for Fig. 4C and PDK1 blot for 5f in IP samples are also poor.

Response: We thank the reviewer for bringing up these concerns. To this end, we have repeated the original Fig. 4c and Fig. 5f and obtained blots with better quality, which are present in the Supplementary Fig. 4c and Supplementary Fig. 6e. We have also carefully examined the Akt pT308 and pS473 blots to ensure their quality.

Reviewer #3 (Remarks to the Author):

I have read the manuscript entitled as “PRMT5-mediated arginine methylation activates AKT kinase to govern tumorigenesis” by Drs Wenjian Gan and coworkers. In this manuscript, authors show that protein arginine methyltransferase 5 (PRMT5) is a methyltransferase of AKT.

1. PRMT5 inhibition (Crisper or shRNA) suppresses AKT activation (Fig 1).
2. AKT rescues cell proliferation in PRMT5-depleted cells (Fig 2).
3. PRMT5 catalyzes symmetric dimethylation of AKT1-R391 (Fig 3).
4. Active form of AKT enhances R391 dimethylation (Fig 4).
5. Absence of R391 methylation blocks AKT activation processes (Fig 5).
6. Absence of R391 methylation attenuates AKT activation and tumorigenesis (fig 6).
7. PRMT5 inhibition sensitizes breast cancer cells to AKT inhibitor and chemotherapeutic agents (Fig 7).

Thus, authors conclude that R391 methylation is an important step for AKT activation and its oncogenic function. In general, studies are logically designed and carefully performed and technically sound, which supported the conclusion. There are few suggestions may need attention.

Response: We thank the reviewer for recognizing the significance and the convincing data of our study, as well as for providing excellent suggestions to guide us to further improve our study.

1. It would be better to add size markers on the side of the immunoblot all the way across the manuscript.

Response: We thank the reviewer for this instructive suggestion. We have added protein size marker in all immunoblots in all figures.

2. AKT is activated by PDK1 on T 308, and mTORC2 on S-473. In the manuscript, it is well documented that PRMT5 can enhance the PDK1 dependent AKT phosphorylation. How mechanistically PRMT5 can enhance the mTORC2 mediated AKT phosphorylation at S-473 (independent from PDK1) may not be well demonstrated. The analysis using TSC1/TSC2 null cells may help to dissect that direction, perhaps.

Response: We thank the reviewer for raising this excellent question and for providing the insightful suggestion. To this end, we have obtained new experimental data described below to demonstrate how PRMT5 promotes AKT-S473 phosphorylation.

- 1) We found that depletion of *PRMT5* weakens the interaction between AKT1 and Sin1, a specific subunit of mTORC2 complex that determines substrate binding to mTORC2 (**revised Supplementary Fig. 6f**). Similar to R25C, R391K mutation also impairs AKT1 binding to Sin1 (**revised Supplementary Fig. 6g**).
- 2) As suggested, we have also examined whether PRMT5 affects AKT activation in Tsc2 null MEFs. TSC1-TSC2 complex can physically associate with mTORC2, but not mTORC1, to promote mTORC2 kinase activity (Huang et al., Mol Cell Biol. 2008). Consistent with the

findings in this paper, we found that AKT-S473 phosphorylation is dramatically suppressed in *Tsc2*^{-/-} MEFs. Moreover, ectopic expression of PRMT5 promotes AKT-S473 phosphorylation in *Tsc2*^{+/+} MEFs, but not *Tsc2*^{-/-} MEFs (*revised Supplementary Fig. 7g*).

- 3) Moreover, a recent study from Dr. Khavari's group (Kovalski et al., Mol Cell. 2019) found that oncogenic Ras mutant directly associates with mTOR and MAPKAP1 (also called Sin1) to promote mTORC2 kinase activity. Following this direction, we have generated MCF7 cells stably expressing NRAS-WT or the oncogenic mutant NRAS-Q61K. Notably, knockout of *PRMT5* reduced AKT-S473 phosphorylation in both cells (*revised Supplementary Fig. 7h*).

Altogether, these results suggest that both mTORC2 kinase activity and interaction with Sin1 are indispensable for AKT-S473 phosphorylation. Therefore, PRMT5-mediated AKT-R391 methylation enhances AKT interaction with Sin1 and subsequently promotes AKT-S473 phosphorylation depending on the mTORC2 kinase activity.

3. Fig 4d. The expression of AKT WT is away higher than R25C (second panel, middle lane), which may enhance the interaction between PRMT5 (top panel, middle lane).

Response: We thank the reviewer for pointing out this concern. We have repeated this experiment by expressing AKT-WT and AKT1-R25C at a comparable level. Consistently, R25C mutation impairs its binding to PRMT5 (*revised Fig. 4d*).

4. In addition to the biochemical data to localize the methylation dependent localization of p-AKT authors are encouraged to show by immunofluorescent study for the intracellular localization of p-AKT will be preferable.

Response: We thank the reviewer for this excellent suggestion. To this end, we have performed immunofluorescent staining of AKT-pS473 and AKT1 in cells expressing AKT-WT or the AKT-R391K mutant. Consistent with the biochemical data, R391K mutation impairs AKT1 and AKT-pS473 membrane localization (*revised Supplementary Fig. 5d and Supplementary Fig. 7f*).

5. Can authors show the interaction between PIP3 with WT-AKT vs R391K-AKT by PIP strip, for example.

Response: We thank the reviewer for this excellent suggestion. We have obtained new experimental data showing that PRMT5-mediated AKT1-R391 methylation promotes AKT1 binding to PIP3. Specifically, we used the PIP3 beads (Echelon Biosciences, #P-B345A) to pulldown AKT1 as described previously (Guo et al., Nat Cell Biol. 2019). Notably, R391K mutation or depletion of *PRMT5* dramatically reduced AKT1 binding to PIP3 (*revised Fig. 5c, d*).

6. Fig. 5 Does PDK1 directly interact with PRMT5 or through PDK1-PIP3 interaction?

Response: We thank the reviewer for raising this question. As suggested, we have ectopically expressed PRMT5 and performed an immunoprecipitation assay to examine the interaction of

PRMT5 and endogenous PDK1 and AKT1. Notably, we could detect only AKT1, but not PDK1, in the HA-PRMT5 immunoprecipitate (*revised Supplementary Fig. 3a*). As we replenished fresh medium for 1 hr before harvesting the cells, it was supposed to be a PIP3-rich condition. Therefore, our results demonstrate that PRMT5 neither directly interacts with PDK1 or through PDK1-PIP3 interaction. However, PRMT5 was proved to interact with AKT1.

7. Fig 3 legend lane 2 c. In vitro “I” seemed to be bold, perhaps.

Response: We thank the reviewer for picking up this typo and have corrected it.

8. Fig 6b. Third panel from the top. WT AKT expression appear to be slightly more than R391K. Can authors confirm expression is even by HA blot, which is tagged to AKT?

Response: We agree with the reviewer that the AKT-WT expression is slightly more than R391K in Fig 6b. As suggested, we have also reblotted these samples with anti-HA antibody and found that the AKT-WT expression is still slightly more than R391K. However, we hope the reviewer agrees with us that this slight difference has minimal impact on the conclusion that R391K mutation suppresses insulin-induced AKT activation as AKT phosphorylation is dramatically different between AKT1-WT and R391K.

REVIEWERS' COMMENTS

Reviewer #1 (Remarks to the Author):

This is a very much improved manuscript, and I thank the authors for their hard work in addressing all of the reviewer comments. I am particularly appreciative that structural modelling of how methylation could be biochemical regulating Akt activation has been included. This is an interesting new insight to the fields of both arginine methylation and Akt signalling. Together, this study should be of interest to many.

I have a few minor points:

I am surprised that over expression of constitutive active Akt does not increase tumour burden or proliferation (Fig 2e and 2f). Is there a reason for this?

In Supp 4b, PRMT5 and AKT interaction increases for the duration of the EGF time course, however Akt-Ser 473 is down at 24 and 32 mins. I understand that total akt levels are also affected, but can WCL bands be quantified and levels of Akt-473 normalised to total Akt to avoid confusion.

Reviewer #2 (Remarks to the Author):

The authors have adequately addressed all my previous concerns, and I have no more questions to ask. The study reveals the novel insight into the role of Akt arginine methylation at R391 by KRMT5 in Akt signaling activation and tumorigenesis. The conclusions are supported by solid data and the study significantly advances our understanding of how oncogenic Akt is regulated by growth factors. It is now stable for publication at Nature Communications.

Reviewer #3 (Remarks to the Author):

I have gone through the revised form of the manuscript entitled as "PRMT5-mediated arginine methylation activates AKT kinase to govern tumorigenesis" by Drs Wenjian Gan and coworkers.

Authors carefully responded my concerns raised in the previous version of the manuscript with satisfactory and convincing manner so that the results presented in the revised form certainly support the conclusion.

Point-by-Point Response to the Reviewers' comments
(Manuscript ID: NCOMMS-20-41840-A)

We thank the reviewer for recognizing the novelty and significance our study and our efforts in this round revision. We also thank the reviewers for the further constructive comments on our revised manuscript. As described in detail below, we have discussed/addressed the additional concerns raising by the reviewer #1. We hope the editor and the reviewers will concur with us that we have satisfactorily addressed all the raised concerns and consequently, publish the paper at *Nature Communications*.

Reviewer #1 (Remarks to the Author):

This is a very much improved manuscript, and I thank the authors for their hard work in addressing all of the reviewer comments. I am particularly appreciative that structural modelling of how methylation could be biochemical regulating Akt activation has been included. This is an interesting new insight to the fields of both arginine methylation and Akt signalling. Together, this study should be of interest to many.

Response: We thank the reviewer for recognizing the novelty and significance our study and our efforts in this round revision. We also appreciate the reviewer for raising additional minor concerns to guide us to further improve our study.

I have a few minor points:

I am surprised that over expression of constitutive active Akt does not increase tumour burden or proliferation (Fig 2e and 2f). Is there a reason for this?

Response: We thank the reviewer for raising this concern. We agree with the reviewer that although overexpression of the myr-AKT increased the tumor burden and proliferation to a certain extent in Fig. 24 and 2f, it is not as dramatic as previous studies. One possibility for this discrepancy is the different expression levels of myr-AKT1 in our study and other studies, which may cause different extent of senescence to affect tumor growth and proliferation. It is well known that hyperactivation of oncogenic pathways may lead to senescence, called oncogene-induced senescence (OIS), which can be overcome by additional genomic hits (Collado Manuel et al. 2005. *Nature*, 436: 642). For example, N-Ras transgenic mice display strong senescence, but loss of Suv39h1 impairs the observed senescence and causes T-cell lymphoma (Braig et al. 2005. *Nature*, 436: 660-665). Notably, constitutively active AKT can also trigger senescence, called AKT-induced senescence (AIS) (Astle et al. 2012. *Oncogene*, 31:1949-1962; Alimonti et al. 2010. *J Clin Invest*, 120:681-93; Jung et al. 2019. *Oncogene*, 38:1639-1650). Therefore, the expression levels of myr-AKT1 may affect the senescence levels. In Fig. 2e and 2f, we infected the cells with lentivirus to stably express myr-AKT1 and then performed mouse xenograft assay, while other studies applied the retrovirus-mediated ectopic expression of myr-AKT1. It is very possible that the lentivirus and retrovirus system may express different levels of myr-AKT, thereby leading to different extent of senescence, tumor growth and proliferation. However, these differences do not affect our conclusion that overexpression of myr-AKT can rescue tumor growth in shPRMT5 cells.

In Supp 4b, PRMT5 and AKT interaction increases for the duration of the EGF time course, however Akt-Ser 473 is down at 24 and 32 mins. I understand that total akt levels are also affected, but can WCL bands be quantified and levels of Akt-473 normalised to total Akt to avoid confusion.

Response: We thank the reviewer for raising this concern. As suggested, we have quantified the levels of Akt-pS473 normalised to total Akt in Supp 4b. After quantification, we found that there is about 1-fold reduction in 24 and 32 min, compared to 8 min. Interestingly, our previous studies also showed that in response to insulin or EGF stimulation, Akt-pS473 is fluctuated during treatment period (Guo et al. 2019. Nat Cell Biol, 21: 226-237; Liu et al. 2013. Nat Cell Biol, 15: 1340-1350). This indicates that except PRMT5 binding and methylating AKT, other mechanisms may also contribute to the Akt-pS473 in response to insulin or EGF, leading to the PRMT5/AKT interaction and Akt-pS473 were not synchronous in this panel.

Reviewer #2 (Remarks to the Author):

The authors have adequately addressed all my previous concerns, and I have no more questions to ask. The study reveals the novel insight into the role of Akt arginine methylation at R391 by KRMT5 in Akt signaling activation and tumorigenesis. The conclusions are supported by solid data and the study significantly advances our understanding of how oncogenic Akt is regulated by growth factors. It is now stable for publication at Nature Communications.

Response: We thank the reviewer for recognizing the novelty and significance our study and for agreeing our manuscript publication at Nature Communications.

Reviewer #3 (Remarks to the Author):

I have gone through the revised form of the manuscript entitled as “PRMT5-mediated arginine methylation activates AKT kinase to govern tumorigenesis” by Drs Wenjian Gan and coworkers. Authors carefully responded my concerns raised in the previous version of the manuscript with satisfactory and convincing manner so that the results presented in the revised form certainly support the conclusion.

Response: We thank the reviewer for agreeing with us that we have addressed all the raising concerns in a satisfactory and convincing manner in this round revision and for agreeing our manuscript publication at Nature Communications.